# From Reward Shaping to Q-Shaping: Achieving Unbiased Learning with LLM-Guided Knowledge

## Abstract

Q-shaping is an extension of Q-value initialization and serves as an alternative to reward shaping for incorporating domain knowledge to accelerate agent training, thereby improving sample efficiency by directly shaping Q-values. This approach is both general and robust across diverse tasks, allowing for immediate impact assessment while guaranteeing optimality. We evaluated Q-shaping across 20 different environments using a large language model (LLM) as the heuristic provider. The results demonstrate that Q-shaping significantly enhances sample efficiency, achieving an **16.87%** average improvement across the 20 tasks compared to the best baseline, and a **226.67%** improvement compared to LLM-based reward shaping methods. These findings establish Q-shaping as an effective and unbiased alternative to conventional reward shaping in reinforcement learning.

## 1 Introduction

Reinforcement learning (RL) can solve complex tasks but often faces sample inefficiency. For example, AlphaGo (Silver et al., 2016) required approximately 4 weeks of training on 50 GPUs, learning from 30 million expert Go game positions to reach a 57% accuracy. Similarly, training a real bipedal soccer robot required $9.0 \times 10^8$ environment steps, amounting to 68 hours of wall-clock time for the full 1v1 agent (Haarnoja et al., 2024). These cases demonstrate the significant computational demands of RL.

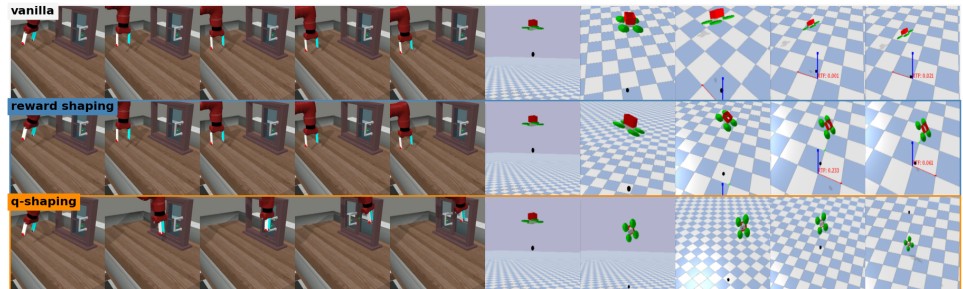

Figure 1: Agent behavior across different algorithms. "Vanilla" refers to traditional RL algorithms, "reward shaping" refers to reward shaping-enhanced RL algorithms, and "Q-shaping" refers to Q-shaping-enhanced RL algorithms. Q-shaping impacts agent behavior quickly, enabling rapid evolution and improvement in the quality of heuristic functions. In contrast, reward shaping requires extensive training time before the impact of the heuristic reward becomes apparent.

To improve efficiency, popular methods include (1) imitation learning, (2) residual reinforcement learning, (3) reward shaping, and (4) Q-value initialization. Yet, each has limitations: imitation learning requires expert data (Garg et al., 2021; Chang et al., 2024; Kostrikov et al., 2020), residual RL needs a well-designed controller (Johannink et al., 2019; Trumpp et al., 2023), and Q-value initialization (Nakamoto et al., 2024) demands precise estimates. Therefore, reward shaping (Xie

et al.; Ma et al., 2023) is the most practical approach, as it avoids the need for expert trajectories or predefined controllers.

Reward shaping methods fall into two main categories: (1) potential-based reward shaping (PBRS) (Ng et al., 1999) and (2) non-potential-based reward shaping (NPBRS) (Ng et al., 1999). PBRS provides state-based heuristic rewards and ensures that optimality is preserved by following the potential function rule, as defined by . NPBRS, on the other hand, refers to reward shaping methods that do not adhere to the potential function rule, and as a result, the learned policy does not guarantee optimality. Additionally, reward shaping methods often suffer from a slow verification process, requiring completion of training to assess the impact of the heuristic reward, which limits their development, as noted by Ma et al. (2023). Lastly, designing high-quality reward functions remains a challenging and often frustrating task for researchers, hindering the adoption of these methods (Ma et al., 2023).

With the growing popularity of large language models (LLMs), LLM-guided reinforcement learning (RL) has emerged as a promising field. This approach leverages the strong understanding capabilities of LLMs to guide RL agents in exploration or policy updates. Existing research has focused on two main areas: LLM-based policy generation and LLM-guided reward design. For example, Chen et al. (2021); Micheli et al. (2022) utilize LLMs to enhance policy decisions, while Kwon et al. (2023); Carta et al. (2023); Ma et al. (2023) employ LLMs to design reward structures. Although these works have improved task success rates, the challenges associated with reward shaping remain unresolved.

In this work, we introduce a novel framework, **Q-shaping**, which leverages domain knowledge from large language models (LLMs) to guide agent exploration. Q-shaping offers two key advantages over reward shaping:

1. **Remain Optimality**: Q-shaping inspires exploration by modifying Q-values during training while ensuring that the agent's optimality remains unaffected upon convergence.
2. **Efficient Heuristic Verification**: Unlike reward shaping methods, which require waiting until the end of training to observe the impact of the reward heuristic, Q-shaping enables experimenters to verify and refine heuristic guidance rapidly during training.

Figure 1 illustrates the agent behavior across different algorithms.

In the "Q-shaping Framework" section, we present theoretical analysis and proofs demonstrating that Q-shaping preserves optimality while using imprecise Q-values to improve exploration and sample efficiency. In the experimental section, we use GPT-4o as a heuristic provider and compare Q-shaping with popular baselines, achieving an average improvement of 16.87% across 20 tasks. Compared to LLM-guided reward shaping methods like T2R (Xie et al.) and Eureka (Ma et al., 2023), Q-shaping achieves up to 226.67% improvement in episodic total rewards while enhancing task success rates.

## 2 RELATED WORK

### 2.1 HEURISTIC REINFORCEMENT LEARNING

There are four common approaches to incorporating domain knowledge into reinforcement learning to enhance sample efficiency: (1) Imitation Learning, (2) Residual Policy, (3) Reward Shaping, and (4) Q-value Initialization.

Imitation Learning requires access to expert trajectories, as demonstrated by works such as GAIL (Ho & Ermon, 2016), where agents learn by mimicking expert behavior. However, the reliance on high-quality expert data limits its applicability in complex tasks. Residual Policy (Johannink et al., 2019) methods involve designing a controller to guide agent actions, but this manual design process restricts their scalability and generality.

Q-value initialization, although promising, often requires precise Q-value estimates to derive an effective policy. For instance, Cal-QL (Nakamoto et al., 2024) employs calibrated Q-values to enhance agent exploration, but these calibrated values still rely on expert knowledge, making Q-value design more challenging than reward shaping. Consequently, few studies have pursued this direction due to the inherent difficulty in obtaining accurate Q-values compared to reward shaping.

Reward shaping directly modifies the reward function to influence agent behavior, improving training efficiency without requiring expert trajectories or manual controller design. This approach has been refined to address diverse learning scenarios, such as in Inverse Reinforcement Learning (IRL) (Ziebart et al., 2008; Wulfmeier et al., 2015; Finn et al., 2016) and Preference-based RL (Christiano et al., 2017; Ibarz et al., 2018; Lee et al., 2021; Park et al., 2022). Additionally, various heuristic techniques have been introduced, including unsupervised auxiliary task rewards (Jaderberg et al., 2016), count-based reward heuristics (Bellemare et al., 2016; Ostrovski et al., 2017), and self-supervised prediction error heuristics (Pathak et al., 2017; Stadie et al., 2015; Oudeyer & Kaplan, 2007).

However, reward shaping often suffers from inaccuracies in the heuristic functions and a slow verification process, which limits its effectiveness in certain applications.

## 2.2 LLM\VLM AGENT

LLMs/VLMs can achieve few-shot or even zero-shot learning in various contexts, as demonstrated by works such as Voyager (Wang et al., 2023), ReAct (Yao et al., 2022), SLINVIT (Zhang et al., 2024),and SwiftSage (Lin et al., 2024).In the field of robotics, VIMA Jiang et al. (2022) employs multimodal learning to enhance agents' comprehension capabilities. Additionally, the use of LLMs for high-level control is becoming a trend in control tasks (Shi et al., 2024; Liu et al., 2023; Ouyang et al., 2024).In web search, interactive agents (Gur et al., 2023; Shaw et al., 2024; Zhou et al., 2023) can be constructed using LLMs/VLMs. Moreover, frameworks have been developed to reduce the impact of hallucinations, such as decision reconsideration (Yao et al., 2024; Long, 2023), self-correction (Shinn et al., 2023; Kim et al., 2024), and observation summarization (Sridhar et al., 2023).

## 2.3 LLM-ENHANCED RL

Relying on the understanding and generation capabilities of large models, LLM-enhanced RL has become a popular field (Du et al., 2023; Carta et al., 2023). Researchers have investigated the diverse roles of large models within reinforcement learning (RL) architectures, including their application in reward design (Kwon et al., 2023; Wu et al., 2024; Carta et al., 2023; Chu et al., 2023; Yu et al., 2023; Ma et al., 2023), information processing (Paischer et al., 2022; 2024; Radford et al., 2021), as a policy generator, and as a generator within large language models (LLMs) (Chen et al., 2021; Micheli et al., 2022; Robine et al., 2023; Chen et al., 2022). While LLM-assisted reward design has improved task success rates (Ma et al., 2023; Xie et al.), it often introduces bias into the original Markov Decision Process (MDP) or fails to provide sufficient guidance for complex tasks. Additionally, the verification process is time-consuming, which slows down the pace of iterative improvements.

## 3 NOTATION

**Markov Decision Processes.** We represent the environment as a Markov Decision Process (MDP) in the standard form: $\mathcal{M} := \langle \mathcal{S}, \mathcal{A}, \mathcal{R}, P, \gamma, \rho \rangle$. Here, $\mathcal{S}$ and $\mathcal{A}$ denote the discrete state and action spaces, respectively. We use $\mathcal{Z} := \mathcal{S} \times \mathcal{A}$ as shorthand for the joint state-action space. The reward function $\mathcal{R}: \mathcal{Z} \to Dist([0,1])$ maps state-action pairs to distributions over the unit interval, while the transition function $P: \mathcal{Z} \to Dist(\mathcal{S})$ maps state-action pairs to distributions over subsequent states. Lastly, $\rho \in Dist(\mathcal{S})$ represents the distribution over initial states. We denote $\mathbf{r}_{\mathcal{M}}$ and $P_{\mathcal{M}}$ as the true reward and transition functions of the environment.

For policy definition, the space of all possible policies is denoted as $\Pi$. A policy $\pi : \mathcal{S} \to \Delta(\mathcal{A})$ defines a conditional distribution over actions given states. A deterministic policy $\mu : \mathcal{S} \to \mathcal{A}$ is a special case of $\pi$, where one action is selected per state with a probability of 1. We define an "activity matrix" $A^{\pi} \in \mathbb{R}^{\mathcal{S} \times \mathcal{Z}}$ for each policy, encoding $\pi$'s state-conditional state-action distribution. Specifically, $A^{\pi}(s, \langle \dot{s}, a \rangle) := \pi(a|s)$ if $s = \dot{s}$, otherwise $A^{\pi}(s, \langle \dot{s}, a \rangle) := 0$. The value function is defined as $v: \Pi \to \mathcal{S} \to \mathbb{R}$ or $q: \Pi \to \mathcal{S} \times \mathcal{A} \to \mathbb{R}$, both with bounded outputs. The terms $\mathbf{q}$ and $\mathbf{v}$ represent discrete matrix representations, where $\mathbf{v}(s)$ and $\mathbf{q}(s, a)$ specifically denote the outputs of an arbitrary value function for a given policy at a particular state or state-action pair.

An *optimal policy* for an MDP $\mathcal{M}$, denoted by $\pi_{\mathcal{M}}^*$, is one that maximizes the expected return under the initial state distribution: $\pi_{\mathcal{M}}^* := \arg\max_\pi \mathbb{E}_\rho[\mathbf{v}_{\mathcal{M}}^\pi]$. The state-wise expected returns of this optimal policy are represented by $\mathbf{v}_{\mathcal{M}}^{\pi_{\mathcal{M}}^*}$. The Bellman consistency equation for the MDP $\mathcal{M}$ at $\mathbf{x}$ is given by $\mathcal{B}_{\mathcal{M}}(\mathbf{x}) := \mathbf{r} + \gamma P\mathbf{x}$. Notably, $(\mathbf{v}_{\mathcal{M}}^\pi)^*$ is the unique vector that satisfies $(\mathbf{v}_{\mathcal{M}}^\pi)^* = A^\pi \mathcal{B}_{\mathcal{M}}((\mathbf{v}_{\mathcal{M}}^\pi)^*)$. We abbreviate $\mathbf{q}^*$ as $(\mathbf{q}_{\mathcal{M}}^{\pi_{\mathcal{M}}^*})^*$ and $\mathbf{q}_\xi^*$ as $(\mathbf{q}_\xi^{\pi_\xi^*})^*$ for some MDP $\xi$.

**Datasets**  We define fundamental concepts essential for fixed-dataset policy optimization. Let $D := \{\langle s, a, r, s'\rangle\}^d$ represent a dataset of $d$ transitions. From this dataset, we can construct a local MDP $\mathcal{D}$ and derive a local optimal Q-value function, denoted as $q_D^*$.

Within the Q-shaping framework, let $\hat{\mathbf{q}}$ denote the Q-function learned from TD estimation and Q-shaping. The LLM outputs are categorized into two types: *goodQ*, which encourages exploration, and *badQ*, which discourages it. Let $G_{LLM} := \{(s, a, Q) \mid Q > 0\}^d$ represent the dataset of $d$ heuristic pairs focused on encouraging agent exploration. Similarly, $B_{LLM} := \{(s, a, Q) \mid Q \leq 0\}^d$ denotes the dataset of $d$ heuristic pairs aimed at preventing exploration. The complete collection of LLM outputs is given by $D_{LLM} := \{G_{LLM}, B_{LLM}\}$.

**Convergence**  An agent is considered to have converged when it reaches 80% of the peak performance. The peak performance is defined as the highest performance achieved by any of the baseline methods.

## 4 Q-SHAPING FRAMEWORK

In the Q-learning framework, an experience buffer $D$ is used to store transitions from the Markov Decision Process (MDP), supporting both online and offline training. To estimate the Q-values for $(s, a)$ pairs, the Temporal-Difference (TD) update method leverages this experience buffer. The Q-function derived from the trained Q-values determines the policy by maximizing $\mathbf{q}(s, \cdot)$, making accurate Q-value estimation crucial for policy quality and effective exploration.

To enhance exploration, Q-shaping integrates both the experience buffer and heuristic guidance from a large language model (LLM) into the Q-value estimation process. The **Heuristic TD Update**, which defines this Q-shaping process, is given by:

$$\hat{\mathbf{q}}^{k+1}(s, a) \leftarrow \begin{cases} \hat{\mathbf{q}}^k(s, a) + \alpha h(s, a), & \text{if } (s, a) \in D_{LLM}^k \setminus \mathcal{D}, \\ \hat{\mathbf{q}}^k(s, a) + \alpha\big(\hat{\mathbf{q}}_{TD}^k(s, a) + h(s, a)\big), & \text{if } (s, a) \in D_{LLM}^k \cap \mathcal{D}. \end{cases}$$

where $\hat{\mathbf{q}}_{TD}^k(s, a)$ represents the temporal-difference (TD) update estimation of $\mathbf{q}(s, a)$ at step $k$, expressed as: $\hat{\mathbf{q}}_{TD}^k(s, a) = r(s, a, s') + \gamma\hat{\mathbf{q}}^k(s, a)$. Here, $D_{LLM}^k$ denotes the set of $(s, a, h(s, a))$ pairs provided by the LLM at iteration $k$.

With this formulation, the **Heuristic Bellman Optimal Operator** can be expressed as:

$$\hat{\mathbf{q}}^{k+1}(s, a) = \mathcal{T}_h \hat{\mathbf{q}}^k(s, a) \tag{1}$$

$$= r(s, a) + \gamma \sum_{s' \in S} P(s'|s, a) \max_{a'} \hat{\mathbf{q}}^k(s', a') + h(s, a), \quad (s, a) \in D_{LLM}^k \cap \mathcal{D}. \tag{2}$$

### 4.1 UNBIASED OPTIMALITY

The Q-value represents a high-level abstraction of an agent's interaction with the environment. It encapsulates the expected cumulative reward by integrating critical elements such as rewards $r$, transition probabilities $P$, states $s$, actions $a$, and the policy $\pi$. Changes in any of these components directly affect the Q-values.

SAC (Haarnoja et al., 2018) and MCTS (Browne et al., 2012) use action-bonus heuristics to enhance training efficiency but risk biasing the learned policy away from optimality. In contrast, Q-shaping, supported by Theorem 1, enhances learning with heuristic guidance while ensuring convergence to the optimal Q-values of the local MDP.

**Theorem 1** (Contraction and Convergence of $\hat{\mathbf{q}}$). *Let $\mathcal{T}_h$ be the heuristic Bellman operator for the sampled MDP $\mathcal{D}$, and let $\gamma \in [0, 1)$ be the discount factor. The operator $\mathcal{T}_h$ satisfies the following contraction property in the metric space $(\mathcal{X}, \|\cdot\|_\infty)$:*

$$\|\mathcal{T}_h(\hat{\mathbf{q}}) - \mathcal{T}_h(\hat{\mathbf{q}}')\|_\infty \leq \gamma\|\hat{\mathbf{q}} - \hat{\mathbf{q}}'\|_\infty,$$

*where $\hat{\mathbf{q}}, \hat{\mathbf{q}}' \in \mathcal{X}$ are any two value functions. Thus, $\mathcal{T}_h$ is a $\gamma$-contraction operator.*

*As a result, repeated applications of the heuristic Bellman operator through the heuristic Temporal Difference (TD) update,*

$$\hat{\mathbf{q}} \leftarrow \mathcal{T}_h(\hat{\mathbf{q}}),$$

*will converge to the unique fixed point $\hat{\mathbf{q}}_\mathcal{D}^*$. Furthermore, since $\hat{\mathbf{q}}$ and $\mathbf{q}$ are updated on the same MDP and Follow Assumption A.2, the following equivalence holds:*

$$\hat{\mathbf{q}}_\mathcal{D}^* = \mathbf{q}_\mathcal{D}^*.$$

*Proof.* See Appendix A.2 □

### 4.2 Utilizing Imprecise Q value Estimation

At the early training stage, the Q-values for different actions are nearly identical, leading the policy to execute actions randomly. To address this, we leverage the LLM's domain knowledge to provide positive Q-values for actions that contribute to task success and negative Q-values for actions that do not. The imprecise Q-values provided by the LLM can be categorized into two types: overestimations and underestimations.

**Underestimation of Non-Optimal Actions** An agent does not need to fully traverse the entire state-action space to identify the optimal trajectory that leads to task success. Therefore, imprecise Q-value estimation can be effectively utilized to guide the agent's exploration.

For instance, consider a scenario where the agent is required to control a robot arm to operate on a drawer located in front of it. In this case, actions such as moving the arm backward or upward are evidently unhelpful in finding the optimal trajectory. Assigning very low Q-values to these non-contributory actions discourages the agent from exploring them, thereby enhancing sample efficiency.

**Overestimation of Near-Optimal Actions** At the initial training phase (iteration step $k = 0$), let action $a$ be assumed to have the highest estimated Q-value for a given state $s$, while $a^*$ denotes the true optimal action. This assumption leads to the inequality $\hat{\mathbf{q}}(s, a^*) < \hat{\mathbf{q}}(s, a) < \mathbf{q}^*(s, a^*)$. Consequently, the agent is predisposed to explore actions around the suboptimal $a$ in its search for states, given that $\mu(s) = \max_a \hat{\mathbf{q}}(s, \cdot) + \epsilon$, where $\epsilon \sim \mathcal{N}(0, \delta^2)$.

However, the number of steps required to discover the optimal action $a^*$ is inherently constrained by the environment and the distance between $a$ and $a^*$. To expedite this exploration process, we introduce an action $a_{LLM}$ suggested by

---

**Algorithm 1** Q-shaping

1: **Require**: Good Q-set $G_{llm}$, Bad Q-set $B_{llm}$ provided by the LLM, RL solver $\mathcal{A}$
2: **Goal**: Compute the average performance over 10 runs
3: **Initialize**: Start 20 agents $\{\text{Agent}_1, \text{Agent}_2, \ldots, \text{Agent}_{20}\}$
4: # for each agent, do:
5: agent.explore(steps = 5000)
6: # Apply Q-shaping and Policy-shaping
7: agent.q_shaping($G_{llm}, B_{llm}$)
8: agent.policy_shaping($G_{llm}, B_{llm}$)
9: # Further exploration
10: agent.explore(steps = 10000)
11: # Synchronize agents
12: agent.wait()
13: # Remove 10 lower-performing agents
14: agent.remove_if_latter()
15: # Continued exploration and training
16: agent.explore_and_train()
17: **Output**: Average performance over 10 runs

---

the LLM, replacing $a$ via Q-shaping guided by the loss function in Equation 3 to enhance sample efficiency. Given the assumption $|a_{LLM} - a^*| < |a - a^*| < \delta$, we can express $\mu(s) = a_{LLM} + \epsilon$. Consequently, the agent has a higher chance of selecting $a^*$, significantly improving the likelihood of identifying the optimal trajectory.

In conclusion, by letting the LLM provide the *goodQ* set and *badQ* set, the agent is guided to prioritize exploring actions suggested by the LLM, thereby enhancing sample efficiency. Over time, as indicated by Hasselt (2010); Fujimoto et al. (2018) and Theorem 1, $\hat{q}$ converges towards the locally optimal Q-function. We now present the theoretical upper bound on the sample complexity required for $\hat{q}$ to converge to $\mathbf{q}^*_{\mathcal{D}}$ for any given MDP $\mathcal{D}$:

**Theorem 2** (Convergence Sample Complexity). *The sample complexity $n$ required for $\hat{q}$ to converge to the local optimal fixed-point $\mathbf{q}^*_D$ with probability $1 - \delta$ is:*

$$n > \mathcal{O}\left(\frac{|S|^2}{2\epsilon^2} \ln \frac{2|S \times A|}{\delta}\right)$$

*Proof.* See proof at A.4. □

Theorem 2 establishes an upper bound on the sample complexity, indicating that the imprecise Q-values provided by the LLM will be corrected within a finite number of steps. Therefore, any heuristic values can be introduced during the early training iterations, and the Q-shaping framework will adapt to inaccurate Q-values over time.

### 4.3 ALGORITHM IMPLEMENTATION

For the implementation of Q-shaping, we employ TD3 (Fujimoto et al., 2018) as the RL solver (backbone) and GPT-4o as the heuristic provider, introducing three additional training phases: (1) Q-Network Shaping (2) Policy-Network Shaping, and (3) High-performance agent selection. Pseudocode 1 outlines the detailed steps of the Q-shaping framework.

**Q-Network Shaping** In the Q-shaping framework, the LLM is tasked with providing a set of $(s, a, Q)$ pairs to guide exploration. This approach is particularly crucial during the early training stage when it is challenging for the agent to independently discover expert trajectories. Traditional RL solvers often require a substantial number of steps to identify the correct path to success, leading to sample inefficiency. The goal of the Q-shaping framework is to leverage the provided $(s, a, Q)$ pairs to accelerate exploration and help the agent quickly identify the optimal path.

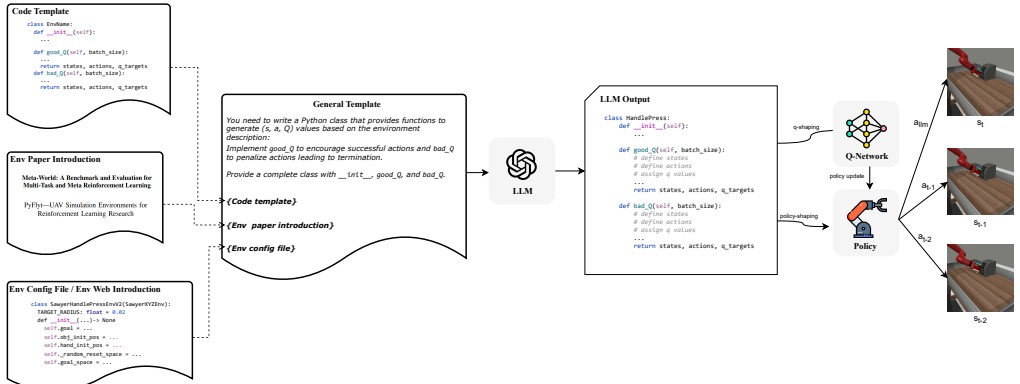

Figure 2: Q-shaping prompt. There is a general code template that specifies the required structure for the generated code. In addition to the template, three key pieces of information are necessary to generate an effective heuristic function: the code template, an introduction to the environment provided in the paper, and the environment configuration file.

To obtain $D_{LLM}$, we construct a general code template as the prompt as illustrated in Figure 2, supplemented by task-specific environment configuration files and a detailed definition of the observation and action spaces within the simulator. Subsequently, we apply the loss function $L_{q-shaping}$ to update the Q-function:

$$L_{q-shaping}(\theta) = E_{(s_i,a_i,Q_i)\sim D_g}(Q_i - \hat{\mathbf{q}}_\theta(s_i, a_i))^2 \tag{3}$$

**Policy-Network Shaping**  In most reinforcement learning (RL) algorithms, the policy is derived from the Q-function, where the policy is optimized to execute actions that maximize the Q-value given a state. The policy update is expressed as: $\mu(s) = \arg\max_a \hat{\mathbf{q}}(s, \cdot)$

While introducing a learning rate and target policy can help stabilize the training process and prevent fluctuations in the policy network, this approach often slows down the convergence speed. To accelerate this adaptation, we introduce a "Policy-Network Shaping" phase designed to allow the policy to quickly align with the good actions and avoid the bad actions provided by the LLM.

The shaping loss function is defined as:

$$L_{policy-shaping} = \lambda_1 \mathbb{E}_{(s,a)\sim G_{LLM}} \left[ \|\mu(s) - a\|^2 \right] - \lambda_2 \mathbb{E}_{(s,a)\sim B_{LLM}} \left[ \|\mu(s) - a\|^2 \right] \quad (4)$$

, where $(s, a) \sim G_{LLM}$ and $(s, a) \sim B_{LLM}$ represent state-action pairs sampled from the LLM-provided *goodQ* and *badQ* sets, respectively, and $\lambda_1$ and $\lambda_2$ are hyperparameters controlling the influence of the LLM-guided shaping.

With this "Policy-Network Shaping" phase, researchers can quickly observe the impact of heuristic values, facilitating the rapid evolution of heuristic quality, ultimately leading to a more efficient exploration process and faster convergence to optimal behavior.

**High-Performance Agent Selection**  With Q-network shaping and policy-network shaping, the policy is initialized to perform actions suggested by the LLM. However, due to the randomness of initial states and the learning process, the shaped agent may still perform poorly and have a chance of failing to learn effectively. To address this, each agent is allowed 10,000 steps to test its performance after the Q-network and policy network initialization. Following this evaluation, weaker agents are removed, and only the top-performing agent continues with the training process.

## 5  EXPERIMENT SETTINGS

We investigate the following **hypotheses** through a series of four experiments:

1. Q-shaping can enhance sample efficiency in reinforcement learning.
2. Q-shaping can adapt to incorrect or hallucinated heuristics while maintaining optimality.
3. Q-shaping outperforms LLM-based reward shaping methods.
4. LLM can design heuristic functions that provide $s, a, Q$ altogether.

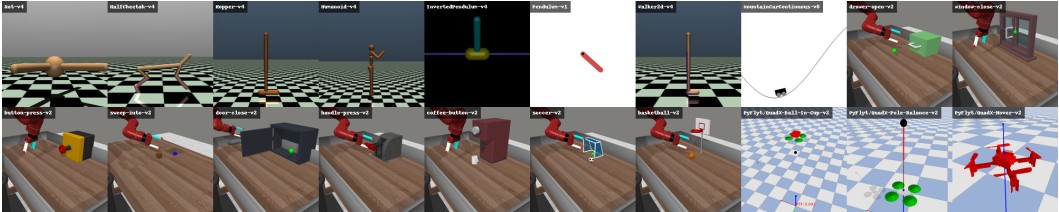

Figure 3: Evaluation environments span a diverse set of robot types and tasks, ranging from simple pendulum systems to humanoid control. The 20 tasks cover a variety of state dimensions, robotic types, and reward structures

To validate these hypotheses, we conducted three primary experiments and two ablation study. GPT-4o served as the heuristic provider, while TD3 was employed as the reinforcement learning (RL) backbone, forming **LLM-TD3**. As illustrated in Figure 3, Q-shaping and various baseline methods were evaluated across 20 distinct tasks involving drones, robotic arms, and other robotic control challenges. Below, we describe the specific experiments and their objectives:

1. **Sample Efficiency Experiment:** This experiment compares Q-shaping with four baseline methods to evaluate its impact on sample efficiency during training.
2. **Comparison with LLM-based Reward Shaping:** Q-shaping, which integrates domain knowledge to assist in agent training, is compared with Text2Reward and Eureka to evaluate its performance relative to existing LLM-based reward shaping approaches.

3. **LLM Quality Evaluation:** Although Q-shaping guarantees optimality, its reliance on LLM-provided heuristics may influence performance. This experiment evaluates the quality of different LLM outputs.

4. **Ablation Study on Q-shaping phases:** Q-shaping introduces three key training phases. This experiment isolates and examines the contribution of each phase to overall performance.

5. **Teachability Experiment:** This experiment evaluates the teachability of different LLMs by analyzing how few interactions can improve code quality and performance.

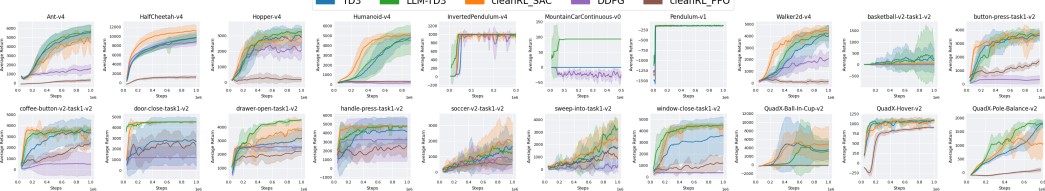

Figure 4: Learning curve comparison of each algorithm across 20 tasks.

**Environments** We evaluate Q-shaping and baselines across 20 distinct environments, including 8 from Gymnasium Classic Control and MuJoCo (Todorov et al., 2012), 9 from MetaWorld (Yu et al., 2020), and 3 from PyFlyt (Tai et al., 2023). Notably, the robotic arm and drone environments used are less commonly studied, making it unlikely that the LLM was pretrained on these specific environments.

**Baselines** For the sample efficiency experiments, we compared Q-shaping against several baseline algorithms, including CleanRL-PPO, CleanRL-SAC (Huang et al., 2022), DDPG (Lillicrap et al., 2015), and TD3 (Fujimoto et al., 2018). When evaluating Q-shaping against other reward shaping methods, we selected Text2Reward and Eureka as baselines. In the LLM-type ablation study, we assessed the performance of different LLMs: o1-Preview, GPT-4o-Mini, Gemini-1.5-Flash (Team et al., 2023), DeepSeek-V2 (DeepSeek-AI et al., 2024), and Yi-Large (Young et al., 2024). - **Text2Reward**: Text2Reward leverages GPT-4 to generate reward functions from natural language task descriptions. In this study, we use provided prompts to describe the MetaWorld tasks, with SAC as the baseline RL algorithm for training policies.

- **Eureka**: Eureka utilizes an evolutionary algorithm to iteratively evolve reward functions based on task performance, refining the reward function over successive generations to improve task success rates. In this work, $K$ (iteration batch size) is set to 8, and $N = 5$ (search iterations) is used. We use GPT-4o as the reward generator and CleanRL-PPO as the backbone reinforcement learning algorithm. The prompts used to generate reward functions are detailed in Appendix B.3.

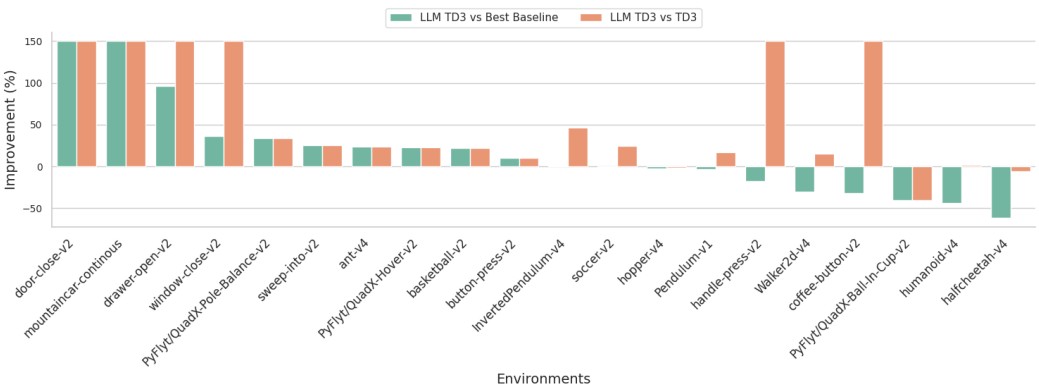

Figure 5: Q-shaping improvement over the best baseline in each environment and its improvement over TD3.

**Metrics**  To evaluate sample efficiency, we measure the number of steps required to reach 80% of peak performance, where peak performance is defined as the highest performance achieved by any baseline agent. For clarity in visualization, improvements exceeding 150% are truncated to 150%.

Each algorithm is tested 10 times, and the average evaluation performance is reported. Evaluations are conducted at intervals of 5,000 steps. During each evaluation, the agent is tested over 10 episodes, and the average episodic return is plotted to form the learning curve.

In this study, we do not specify a fixed seed for each run. Using a fixed seed results in a unique initial state when the environment is reset, which simplifies learning and makes it challenging to accurately verify the effectiveness and generalization capabilities of each algorithm.

## 6  RESULTS AND ANALYSIS

**Q-Shaping Outperforms Best Baseline by an Average of 16.87% Across 20 Tasks**  As shown in Figure 5 and Figure 4, Q-shaping demonstrated a notable improvement over both the best baseline and TD3 across 20 tasks. On average, Q-shaping improves performance by 16.87% compared to the best baseline and by 55.39% compared to TD3, highlighting its effectiveness in enhancing sample efficiency and task performance. This supports **H1**.

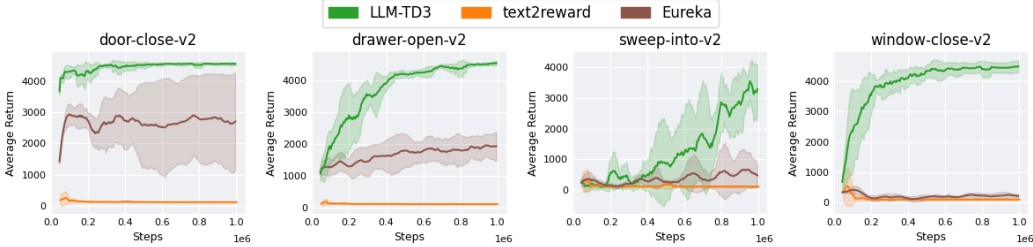

Figure 6: Learning curve comparison between Q-shaping and LLM-based reward shaping methods. The evaluation was conducted on four Meta-World environments: *door-close*, *drawer-open*, *window-close*, and *sweep-into*, with peak performance serving as the basis for comparison.

**Q-Shaping Outperforms LLM-Based Reward Shaping Methods by 226.67%**  Q-shaping achieved substantial improvements over both the Eureka and T2R baselines, as shown in Figure 6. The comparison is based on peak performance across the evaluated Meta-World environments.

Compared to the best baseline, LLM-TD3 improved by 5.16% in the *door-close* task, 81.89% in *drawer-open*, 715.67% in *window-close*, and 103.96% in *sweep-into*, resulting in an average peak performance improvement of 226.67%.

Table 1: Additional training steps required to derive effective heuristic functions for LLM-TD3 and Eureka across four Meta-World environments.

| Algorithm | door-close-v2 | drawer-open-v2 | sweep-into-v2 | window-close-v2 |
|---|---|---|---|---|
| Eureka | $8 \times 10^6$ | $8 \times 10^6$ | $8 \times 10^6$ | $8 \times 10^6$ |
| LLM-TD3 | $1.5 \times 10^3$ | $2 \times 10^3$ | $3 \times 10^3$ | $2 \times 10^3$ |

LLM-based reward shaping methods, though capable of improving task success rates (Ma et al., 2023; Xie et al.), often bias optimality and, as shown in Table 1, require substantial time to evaluate the effectiveness of reward heuristics. In contrast, Q-shaping achieves a 226.67% improvement over the best LLM-based reward shaping methods and requires only a few steps to validate the heuristic function. This supports **H2** and **H3**.

**Most LLMs Can Provide Correct Heuristic Functions**  We evaluated the quality of LLM-generated heuristic functions from five perspectives: (1) adherence to the required code template, (2) correctness of the assigned Q-values, (3) accuracy of the state-action dimension, (4) completeness of the generated code, and (5)

Table 2: Evaluation of LLM Quality in Outputting Heuristic Values

| Metric | o1-Preview | GPT-4o | Gemini | DeepSeek-V2.5 | yi-large |
|---|---|---|---|---|---|
| Template Adherence (%) | 100.0 | 100.0 | 40.0 | 100.0 | 100.0 |
| Correct Q-values (%) | 100.0 | 100.0 | 60.0 | 100.0 | 100.0 |
| Correct State-Action Dim (%) | 100.0 | 100.0 | 80.0 | 100.0 | 100.0 |
| Code Completeness (%) | 100.0 | 100.0 | 20.0 | 100.0 | 100.0 |
| Bug-Free (%) | 100.0 | 100.0 | 20.0 | 100.0 | 100.0 |
| Average (%) | 100.0 | 100.0 | 44.0 | 100.0 | 100.0 |

presence of bugs in the generated code. Each LLM was prompted 10 times with the same request, and we quantified their performance using a correctness rate across these metrics.

Table 3: Ablation Study on Additional Training Phases. The study evaluates the impact of three key training phases—Q-Network Shaping, Policy-Network Shaping, and Agent Selection—across four Meta-World environments: *door-close*, *drawer-open*, *window-close*, and *sweep-into*. Effectiveness is measured by convergence steps ($\downarrow$), with "Failed" indicating algorithms that did not reach the convergence threshold within $10^6$ steps.

| Phase | | | Environment ($\downarrow$) | | | |
|---|---|---|---|---|---|---|
| Q-shaping | Policy-shaping | Selection | door-close-v2 | drawer-open-v2 | sweep-into-v2 | window-close-v2 |
| $\times$ | $\times$ | $\times$ | Failed | Failed | Failed | 759999 |
| $\checkmark$ | $\times$ | $\times$ | Failed | 310000 | Failed | 570000 |
| $\times$ | $\checkmark$ | $\times$ | 30000 | 340000 | Failed | 215000 |
| $\checkmark$ | $\checkmark$ | $\times$ | 30000 | 275000 | 860000 | 195000 |
| $\checkmark$ | $\checkmark$ | $\checkmark$ | 25000 | 265000 | 790000 | 165000 |

Correctness of the assigned Q-values means that state-action pairs $(s, a)$ from the LLM-generated *goodQ* set must be assigned Q-values greater than zero, while those from the *badQ* set must be assigned Q-values less than or equal to zero.

The results, as shown in Table 2, indicate that most LLMs, including o1-Preview, GPT-4o, DeepSeek-V2.5, and yi-large, provided correct heuristic functions with a 100% success rate across all evaluation metrics. However, Gemini exhibited poorer performance, achieving only 44% on average. This supports **H4**.

**Each Training Phase Enhances Sample Efficiency**  As shown in Table 3, each training phase enhances sample efficiency. Q-Network shaping and policy-network shaping together result in substantial performance gains for TD3. Additionally, the agent selection phase helps by eliminating agents that fail to explore effective trajectories in the early stages of training, providing a slight improvement in average sample efficiency.

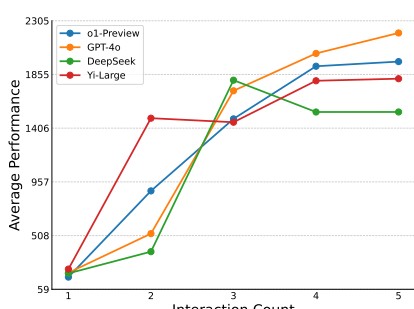

Figure 7: Teachability of different LLMs. The x-axis represents the number of interactions, while the y-axis shows the average performance across four tasks: *Door-Close*, *Drawer-Open*, *Sweep-Into*, and *Window-Close*.

**Few Interactions Significantly Improve Code Quality**  Figure 7 illustrates the teachability of LLMs within the Q-shaping framework. Remarkably, all models achieved high performance within just 3 to 4 interactions, suggesting that the primary issue with the initial generated code lies in parameter tuning rather than structural flaws.

## 7 CONCLUSION

We propose Q-shaping, an alternative framework that integrates domain knowledge to enhance sample efficiency in reinforcement learning. In contrast to traditional reward shaping, Q-shaping offers two key advantages: (1) it preserves optimality, and (2) it allows for rapid verification of the agent's behavior. These features enable experimenters or LLMs to iteratively refine the quality of heuristic values without concern for the potential negative impact of poorly designed heuristics. Experimental results demonstrate that Q-shaping significantly improves sample efficiency and outperforms LLM-guided reward shaping methods across various tasks.

We hope this work encourages further research into advanced techniques that leverage LLM outputs to guide and enhance the search process in reinforcement learning.

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

# A APPENDIX

## A.1 ADDITIONAL NOTATION

**Datasets.** In practise, a batch of data will be sampled from a distribution $\Phi : Dist(\mathcal{Z})$, which is the collected local MDP. A batch $D'$ containing $d$ tuples $\langle s, a, r, s' \rangle$ is sampled as $D' \sim \Phi_d$, where pairs $\langle s, a \rangle$ are drawn from $\Phi$, and rewards $r$ and subsequent states $s'$ are sampled independently from the reward function $\mathcal{R}_D(\cdot|\langle s, a \rangle)$ and the transition function $P_D(\cdot|\langle s, a \rangle)$, respectively.

Given a dataset or batch $D$, we denote $D(\langle s, a \rangle)$ as the multi-set of all $\langle r, s' \rangle$ pairs, and use $\ddot{\mathbf{n}}_D \in \mathbb{R}^{|\mathcal{Z}|}$ to denote the count vector, where $\ddot{\mathbf{n}}_D(\langle s, a \rangle) := |D(s, a)|$. We define the empirical reward vector as $\mathbf{r}_D(\langle s, a \rangle) := \sum_{r, s' \in D(\langle s, a \rangle)} \frac{r}{|D(\langle s, a \rangle)|}$ and empirical transition matrix as $P_D(s'|\langle s, a \rangle) := \sum_{r, \dot{s}' \in D(\langle s, a \rangle)} \frac{\mathbb{I}(\dot{s}' = s')}{|D(\langle s, a \rangle)|}$ for all state-action pairs with $\ddot{\mathbf{n}}_D(\langle s, a \rangle) > 0$. For state-action pairs where $\ddot{\mathbf{n}}_D(\langle s, a \rangle) = 0$, the maximum-likelihood estimates of reward and transition cannot be clearly defined, so they remain unspecified. The bounds hold no matter how these values are chosen, so long as $\mathbf{r}_D$ is bounded and $P_D$ is stochastic. The empirical policy of a dataset $D$ is defined as $\hat{\pi}_D(a|s) := \frac{|D(\langle s, a \rangle)|}{|D(\langle s, \cdot \rangle)|}$ except where $\ddot{\mathbf{n}}_D(\langle s, a \rangle) = 0$, where it can similarly be any valid action distribution. The empirical visitation distribution of a dataset $D$ is computed analogously to the regular visitation distribution but uses $P_D$ in place of $P$. Thus it's given by $\frac{1}{1-\gamma} (I - \gamma A^\pi P_D)^{-1}$.

**Lemma 1** (Decomposition). *For any MDP $\xi$ and policy $\pi$, consider the Bellman fixed-point equation given by, let $(\mathbf{v}_\xi^\pi)^*$ be defined as the unique value vector such that $(\mathbf{v}_\xi^\pi)^* = A^\pi(\mathbf{r}_\xi + \gamma P_\xi(\mathbf{v}_\xi^\pi)^*)$, and let $\mathbf{v}$ be any other value vector. Assume that $\pi(a|s) = 1$ if $a = \arg\max_a (\mathbf{q}_\xi^\pi)^*(s, a)$, otherwise $\pi(a|s) = 0$. We have:*

$$|\mathbf{q}_\xi^*(s, \mu(s)) - \mathbf{q}(s, \mu(s))| = |((I - \gamma A^\pi P_\xi)^{-1}(A^\pi(\mathbf{r}_\xi + \gamma P_\xi \mathbf{v}) - \mathbf{v}))(s)| \tag{5}$$

*Proof.*

$$
\begin{aligned}
A^\pi(\mathbf{r}_\xi + \gamma P_\xi \mathbf{v}) - \mathbf{v} &= A^\pi(\mathbf{r}_\xi + \gamma P_\xi \mathbf{v}) - (\mathbf{v}_\xi^\pi)^* + (\mathbf{v}_\xi^\pi)^* - \mathbf{v} \\
&= A^\pi(\mathbf{r}_\xi + \gamma P_\xi \mathbf{v}) - A^\pi(\mathbf{r}_\xi + \gamma P_\xi(\mathbf{v}_\xi^\pi)^*) + (\mathbf{v}_\xi^\pi)^* - \mathbf{v} \\
&= \gamma A^\pi P_\xi(\mathbf{v} - (\mathbf{v}_\xi^\pi)^*) + ((\mathbf{v}_\xi^\pi)^* - \mathbf{v}) \\
&= ((\mathbf{v}_\xi^\pi)^* - \mathbf{v}) - \gamma A^\pi P_\xi((\mathbf{v}_\xi^\pi)^* - \mathbf{v}) \\
&= (I - \gamma A^\pi P_\xi)((\mathbf{v}_\xi^\pi)^* - \mathbf{v})
\end{aligned}
$$

Note that $(\mathbf{v}_\xi^\pi)^* = A^\pi(\mathbf{q}_\xi^\pi)^*$, After we expand the value function we have:

$$
\begin{aligned}
(I - \gamma A^\pi P_\xi)^{-1}\big(A^\pi(\mathbf{r}_\xi + \gamma P_\xi \mathbf{v})\big) &= A^\pi(\mathbf{q}_\xi^\pi)^* - \mathbf{v} \\
&= A^\pi(\mathbf{q}_\xi^\pi)^* - A\mathbf{q}
\end{aligned}
$$

By indexing at $\langle s, \mu(s) \rangle$, we have:

$$|\mathbf{q}_\xi^*(s, \mu(s)) - \mathbf{q}(s, \mu(s))| = |((I - \gamma A^\pi P_\xi)^{-1}(A^\pi(\mathbf{r}_\xi + \gamma P_\xi \mathbf{v}) - \mathbf{v}))(s)|$$

$\square$

**Lemma 2** (Convergence Bound). *Since that $s'$ and $r$ are sampled independently and identically distributed (iid) from $P_D(\cdot|s, a)$ and $R_D(\cdot|s, a)$ respectively. Let $D'$ denotes the batch of data sample from $D$. Then, with probability at least $1 - \delta$, we have:*

$$|\mathbf{q}_{D'}^*(s, \mu(s)) - \hat{\mathbf{q}}(s, \mu(s))| \leq \left( \sqrt{\frac{1}{2} \ln \frac{2|\mathcal{S} \times \mathcal{A}|}{\delta}} \right) \sum_{s'} \nu_{D'}(s'|s_0 = s) \frac{1}{\sqrt{\ddot{\mathbf{n}}_{D'}(\langle s', \mu(s') \rangle)}}$$

*Proof.* See proof at A.3 $\square$

## A.2 PROOF OF THEOREM 1

**Assumption A.1.** *The heuristic $h(s, a)$ provided by the LLM does not change with the training steps, i.e., $h^k(s, a) = h(s, a)$ for all $k = 0, 1, 2, \ldots$.*

**Assumption A.2.** *The heuristic $h(s, a)$ is only used during the initial training steps and is removed after some step $k_0$, i.e., for all training steps $k \geq k_0$, the heuristic term is not provided.*

Note that $\mathbf{q}$ is the matrix representation of the Q function. In the proof of this section, we use a more general $Q : \mathbb{R}^{\mathcal{Z}} \to \mathbb{R}$ to represent the Q function. The heuristic TD update for $\hat{Q}$ iteration is:

$$\hat{Q}^{k+1}(s, a) = (1 - \alpha)\hat{Q}^k(s, a) + \alpha \left( r(s, a) + \gamma \sum_{s' \in S} P(s'|s, a) \max_{a'} \hat{Q}(s', a') + h(s, a) \right)$$

.

We can define a **Bellman optimal operator** $\mathcal{T}_h$ based on the heuristic TD update as follows:

$$\hat{Q}^{k+1}(s, a) = \mathcal{T}_h \hat{Q}^k = r(s, a) + \gamma \sum_{s' \in S} P(s'|s, a) \max_{a'} \hat{Q}^k(s', a') + h(s, a)$$

.

Suppose training framework Q-shaping satisfies assumption A.1. Then we prove that the Bellman optimal operator $\mathcal{T}_h$ is $\gamma$-contraction operator on $\hat{Q}$:

$$\|\mathcal{T}_h \hat{Q} - \mathcal{T}_h \hat{Q}'\|_\infty = \gamma \max_{s, a \in \mathcal{S}, \mathcal{A}} | \sum_{s'} P(s'|s, a)[\max_{a'} \hat{Q}(s', a') - \max_{a'} \hat{Q}'(s', a')]|$$

$$\leq \gamma \max_{s, a \in \mathcal{S}, \mathcal{A}} |\max_{s'}| \left( \max_{a'} \hat{Q}(s', a') - \max_{a'} \hat{Q}'(s', a') \right) ||$$

$$= \gamma \|\hat{Q} - \hat{Q}'\|_\infty$$

The optimal Q-function for the new update formula, without assumption A.2, is defined as:

$$\hat{Q}^*(s, a) = r(s, a) + \gamma \sum_{s' \in S} P(s'|s, a) \max_{a'} \hat{Q}^*(s', a') + h(s, a)$$

.

$\mathcal{T}_h$ is a $\gamma$-contraction operator on $\hat{Q}$. This means that as the number of iterations k increases, $\hat{Q}$ will approach the heuristic fixed point, which is biased. Under assumption A.2, the heuristic TD update will degenerate into the TD update. Without the influence of the heuristic term, the Q-values will be estimated solely from the local MDP $\mathcal{D}$.

Next, we prove that the converged heuristic-guided Q function is equivalent to the traditional Q function. Define the following:

$$\Theta_H \text{ denotes the set of terminal states,}$$

$$\Theta_{H-1} \text{ denotes the set of states one step before the terminal,}$$

$$\vdots$$

$$\Theta_1 \text{ denotes the set of states at the initial step.}$$

For all $s \in \Theta_{H-1}$ and some action $a$, it is clear that $\hat{Q}^*(s, a) = Q^*(s, a)$, because:

$$Q^*(s, a)|_{s \in \Theta_{H-1}} = \hat{Q}^*(s, a)|_{s \in \Theta_{H-1}} = r(s, a) + \gamma \sum_{s' \in \Theta_H} \mathbf{1}_{s \in \Theta_H} \max_{a'} Q^*(s', a') = r(s, a)$$

For all $s \in \Theta_{H-2}$ and some action $a$, we have:

$$\hat{Q}^*(s, a)|_{s \in \Theta_{H-2}} = r(s, a) + \gamma \sum_{s' \in \Theta_{H-1}} P(s'|s, a) \max_{a'} \hat{Q}^*(s', a')$$

$$= r(s, a) + \gamma \sum_{s' \in \Theta_{H-1}} P(s'|s, a) \max_{a'} Q^*(s', a')$$

$$= Q^*(s, a)|_{s \in \Theta_{H-2}}$$

.

With sufficient iterations, we have: $\hat{Q}^* = Q^*$ . Specifically, we have: $\mathbf{q}^* = \hat{\mathbf{q}}^*$ for some MDP $\mathcal{D}$.

### A.3 PROOF OF LEMMA 2

Let $D'$ be a batch of data, and $D$ denotes the replay buffer, consider that for any $\langle s, a \rangle$, the expression $\mathbf{r}_{D'}(\langle s, a \rangle) + \gamma P_{D'}(\langle s, a \rangle)\mathbf{v}^\pi$ can be equivalently expressed as an expectation of random variables,

$$\mathbf{r}_{D'}(\langle s, a \rangle) + \gamma P_{D'}(\langle s, a \rangle)\mathbf{v} = \frac{1}{\ddot{\mathbf{n}}_{D'}(\langle s, a \rangle)} \sum_{r,s' \in D'(\langle s,a \rangle)} r + \gamma \mathbf{v}(s')$$

each with expected value:

$$\mathbb{E}_{r,s' \in D'(\langle s,a \rangle)}[r + \gamma \mathbf{v}(s')] = \mathbb{E}_{\substack{r \sim \mathcal{R}_D(\cdot | \langle s,a \rangle) \\ s' \sim P_D(\cdot | \langle s,a \rangle)}}[r + \gamma \mathbf{v}(s')] = [\mathbf{r}_D + \gamma P_D \mathbf{v}](\langle s, a \rangle).$$

Hoeffding's inequality indicates that the mean of bounded random variables will approximate their expected values with high probability. By applying Hoeffding's inequality to each element in the $|\mathcal{S} \times \mathcal{A}|$ state-action space and employing a union bound, we establish that with probability at least $1 - \delta$,

$$|(\mathbf{r}_D + \gamma P_D \mathbf{v}) - (\mathbf{r}_{D'} + \gamma P_{D'} \mathbf{v})| \leq \frac{1}{1-\gamma} \sqrt{\frac{1}{2} \ln \frac{2|\mathcal{S} \times \mathcal{A}|}{\delta}} \ddot{\mathbf{n}}_{D'}^{-1}$$

We can left-multiply $A^\pi$ and rearrange to get:

$$|A^\pi(\mathbf{r}_D + \gamma P_D \mathbf{v}) - A^\pi (\mathbf{r}_{D'} + \gamma P_{D'} \mathbf{v})| \leq \left( \frac{1}{1-\gamma} \sqrt{\frac{1}{2} \ln \frac{2|\mathcal{S} \times \mathcal{A}|}{\delta}} \right) A^\pi \ddot{\mathbf{n}}_{D'}^{-\frac{1}{2}}$$

then we left-multiply the discounted visitation of $\pi$:

$$|\left(I - \gamma A^\pi P_{D'}\right)^{-1} [A^\pi(\mathbf{r}_D + \gamma P_D \mathbf{v}) - A^\pi (\mathbf{r}_{D'} + \gamma P_{D'} \mathbf{v})]| \leq \left( \frac{1}{1-\gamma} \sqrt{\frac{1}{2} \ln \frac{2|\mathcal{S} \times \mathcal{A}|}{\delta}} \right) \left(I - \gamma A^\pi P_{D'}\right)^{-1} A^\pi \ddot{\mathbf{n}}_{D'}^{-\frac{1}{2}}$$

This matrix: $\left(I - \gamma A^\pi P_{D'}\right)^{-1} A^\pi \ddot{\mathbf{n}}_{D'}^{-\frac{1}{2}}$, belongs to the space $\mathbb{R}^{|S|}$. By indexing at state $s$, we obtain:

$$\left(I - \gamma A^\pi P_{D'}\right)^{-1} A^\pi \ddot{\mathbf{n}}_{D'}^{-\frac{1}{2}}(s) = (1-\gamma) \sum_{s'} \nu(s'|s_0 = s) \frac{1}{\sqrt{N_{D'}(\langle s, \mu(s) \rangle)}}$$

Finally, by integrate these terms together we have the bound on Lemma 2:

$$|\mathbf{q}_{D'}^*(s, \mu(s)) - \mathbf{q}(s, \mu(s))| \leq \left( \sqrt{\frac{1}{2} \ln \frac{2|\mathcal{S} \times \mathcal{A}|}{\delta}} \right) \sum_{s'} \nu(s'|s_0 = s) \frac{1}{\sqrt{N_{D'}(\langle s', \mu(s') \rangle)}}$$

Given that this inequality is universally applicable to any $\mathbf{q}$, and acknowledging that the heuristic term $\mathbf{h}$ supplied by the LLM serves as a constant within the temporal-difference (TD) update mechanism of the Q-function, it follows that:

$$|\mathbf{q}_{D'}^*(s, \mu(s)) - \hat{\mathbf{q}}(s, \mu(s))| = |\mathbf{q}_{D'}^*(s, \mu(s)) - \mathbf{q}(s, \mu(s)) - \mathbf{h}(s, \mu(s))|$$

$$\leq \left( \sqrt{\frac{1}{2} \ln \frac{2|\mathcal{S} \times \mathcal{A}|}{\delta}} \right) \sum_{s'} \nu(s'|s_0 = s) \frac{1}{\sqrt{N_{D'}(\langle s', \mu(s') \rangle)}}$$

### A.4 PROOF OF THEOREM 2

To get the sample complexity of convergence. By Lemma 2,we have:

$$
\begin{aligned}
|\mathbf{q}^*_{D'}(s,\mu(s)) - \hat{\mathbf{q}}(s,\mu(s))| &\leq \left(\sqrt{\frac{1}{2}\ln\frac{2|\mathcal{S}\times\mathcal{A}|}{\delta}}\right)\sum_{s'}\nu(s'|s_0=s)\frac{1}{\sqrt{N_{D'}(\langle s',\mu(s')\rangle)}} \\
&= \left(\sqrt{\frac{1}{2}\ln\frac{2|\mathcal{S}\times\mathcal{A}|}{\delta}}\right)\sum_{s'}\sqrt{\nu(s'|s_0=s)}\frac{\sqrt{\nu(s'|s_0=s)}}{\sqrt{nd_{D'}(s,a)}} \\
&\qquad\qquad\qquad ( d_{D'}(s,a) = \tfrac{N_{D'}(\langle s,a\rangle)}{|D'|}) \\
&= \left(\sqrt{\frac{1}{2}\ln\frac{2|\mathcal{S}\times\mathcal{A}|}{\delta}}\right)\sum_{s'}\sqrt{d_{D'}(s,\mu(s))}\frac{\sqrt{d_{D'}(s,\mu(s))}}{\sqrt{nd_{D'}(s,a)}} \\
&\qquad\qquad\qquad ( \nu(s)\pi(\mu(s)|s) \approx d_{D'}(s,\mu(s))) \\
&\leq \frac{\left(\sqrt{\frac{1}{2}\ln\frac{2|\mathcal{S}\times\mathcal{A}|}{\delta}}\right)}{\sqrt{n}}\sum_{s'}\sqrt{d_{D'}(s',\mu(s))} \\
&\leq \left(\sqrt{\frac{1}{2}\ln\frac{2|\mathcal{S}\times\mathcal{A}|}{\delta}}\right)\frac{|S|}{\sqrt{n}}
\end{aligned}
$$

Since $D'$ is sampled iid from replay buffer $D$, Then, when $n > \mathcal{O}\left(\frac{|S|^2}{2\epsilon^2}\ln\frac{2|S\times A|}{\delta}\right)$,we have $|\mathbf{q}^*_D(s,\mu(s)) - \mathbf{q}^*(s,\mu(s))| \leq \epsilon$.

## B EXPERIMENT DETAILS

### B.1 Q-SHAPING DETAILS

In our experiments, we utilized "gpt-4o" as the language model to provide heuristic Q-values, thereby accelerating the exploration process in the **LLM-TD3** algorithm. The experiments were conducted on a host equipped with a 48-core CPU, 24 GB of GPU memory, and 120 GB of RAM. For complex tasks, the agent took approximately 2 to 4 hours to converge, whereas for simpler tasks, convergence was achieved within 10 to 30 minutes. Table 4 provides a detailed description of the experimental environment.

Table 4: Experimental Environment

| Resource | Specification |
| --- | --- |
| CPU | 48-core Intel Xeon E5-2666 v4 |
| GPU | NVIDIA GeForce RTX 4090 (24 GB) |
| RAM | 118.1 GB |
| Convergence Time (Complex Tasks) | 2-4 hours |
| Convergence Time (Simple Tasks) | 10-30 minutes |

**Hyperparameters**  LLM-TD3 is built on top of TD3, and doesn't require parameter tuning. In the baseline implementation, TD3's hyperparameters are also fixed for comparison. The hyperparameters of LLM-TD3 are detailed in Table 5. Table 6 displays the convergence line for each environment.

Table 5: Hyperparameters of LLM-TD3

| Hyperparameter | Value |
|---|---|
| LLM Type | gpt-4o |
| Start Timesteps | 5000 |
| Evaluation Frequency | 5,000 |
| Exploration Noise (Std) | 0.1 |
| Batch Size | 256 |
| Discount Factor $\gamma$ | 0.99 |
| Target Network Update Rate (Tau) | 0.005 |
| Policy Noise | 0.2 |
| Noise Clip | 0.5 |
| Policy Update Frequency | 2 |
| $\lambda_1,\lambda_2$ | 100,10 |
| Hidden Layer Size | 512 (10,240 for Humanoid) |

Table 6: Convergence Line for Each Environment

| Environment | Convergence Line |
|---|---|
| Ant-v4 | 4480 |
| HalfCheetah-v4 | 8800 |
| Hopper-v4 | 2560 |
| Humanoid-v4 | 4000 |
| InvertedPendulum-v4 | 800 |
| Pendulum-v1 | -200 |
| Walker2D-v4 | 3700 |
| MountainCarContinuous | 0.1 |
| Drawer-Open-Task1 | 3200 |
| Window-Close-Task1 | 3200 |
| Button-Press-Task1 | 3200 |
| Sweep-Into-Task1 | 2800 |
| Door-Close-Task1 | 3200 |
| Handle-Press-Task1 | 3200 |
| Basketball-V2-Task1 | 360 |
| Coffee-Button-V2-Task1 | 2960 |
| Soccer-V2-Task1 | 1600 |
| PyFlyt/QuadX-Ball-In-Cup-V2 | 3840 |
| PyFlyt/QuadX-Pole-Balance-V2 | 1600 |
| PyFlyt/QuadX-Hover-V2 | 880 |

## B.2 BASELINE DETAILS

We use table 7 to list the open source repositories of the algorithms used in the experiment, Figure 8 to present the hyperparameters of cleanRL_SAC, and Figure 9 to present the hyperparameters of cleanRL_PPO.

Table 7: Baseline Code Source

| Algorithm | Code Repository |
|---|---|
| cleanRL_PPO | https://github.com/vwxyzjn/cleanrl |
| TD3 | https://github.com/sfujim/TD3 |
| DDPG | https://github.com/sfujim/TD3 |
| cleanRL_SAC | https://github.com/vwxyzjn/cleanrl |

Table 8: Hyperparameters of SAC

| Hyperparameter | Value |
|---|---|
| Critic Learning Rate | 3e-3 |
| Actor Learning Rate | 3e-4 |
| Entropy Target | $-\dim(\mathcal{A})$ |
| Policy Update Frequency | 1 |
| Reward Scale | $\frac{1}{8}, 1$ |
| Hidden Layer Size | 128 |

Table 9: Key Hyperparameters of PPO

| Hyperparameter | Value |
|---|---|
| Learning Rate | 3e-4 |
| Num Steps | 2048 |
| Total Timesteps | 1e6 |
| Gamma (Discount Factor) | 0.99 |
| GAE Lambda | 0.95 |
| Clip Coefficient | 0.2 |

### B.3 DETAILS OF IMPLEMENTING LLM-BASED REWARD SHAPING METHODS

In this experiment, we evaluate Q-shaping against Text2Reward (T2R) (Xie et al.) and Eureka (Ma et al., 2023) to compare LLM-based reward shaping approaches.

**Text2Reward (T2R):** Text2Reward is a framework designed to address the challenge of reward shaping in reinforcement learning by automating the generation of dense, interpretable reward codes using large language models (LLMs). This method demonstrates effectiveness across various robotic and locomotion tasks, achieving success rates comparable to or exceeding those obtained with expert-designed reward codes (Xie et al.). In our experiment, we implement T2R using the provided prompt available at GitHub link and Soft Actor-Critic (SAC) as the RL backbone. The hyperparameters listed in Table 10 are used for the implementation of SAC in the Text2Reward experiment.

Table 10: Key Hyperparameters for SAC Implementation

| **Hyperparameter** | **Value** |
|---|---|
| Batch Size | 512 |
| Policy Network Architecture | [256, 256, 256] |
| Discount Factor ($\gamma$) | 0.99 |
| Learning Rate | 0.0003 |
| Soft Update Coefficient ($\tau$) | 0.005 |
| Learning Starts (steps) | 25,000 |
| Entropy Coefficient ($\alpha$) | auto_0.1 |

**Eureka:**

1. Eureka is a reward design algorithm that leverages the capabilities of LLMs for evolutionary optimization of reward functions. It uses the environment code as context, generating executable reward functions in a zero-shot manner, and iteratively improves them through reflection-based feedback and evolutionary search. Eureka's robust framework has been validated across a wide range of RL tasks, outperforming expert-designed rewards in many scenarios (Ma et al., 2023).

2. Eureka is originally designed to operate within the Isaac Gym simulator, adaptations were necessary for our experiments to integrate Eureka's functionality with our environment. Specifically, the prompt for Eureka was tailored into two configurations: one for initial code generation and another for refining the code based on feedback. These prompts are detailed in `Eureka Prompt 1: Code Generation` and `Eureka Prompt 1: Reflection`. The first prompt facilitates the generation of foundational reward programs, while the second focuses on optimizing these codes iteratively to align better with experimental objectives.

In our implementation of Eureka, we configured the iterative batch size ($K$) to 8 and the search iterations ($N$) to 5. Table 11 summarizes the results of each evolutionary iteration. It shows agent performance at each run and the improvement of each evolution.

---

**Eureka Prompt 1: Code Generation**

You are a reward engineer trying to write reward functions to solve reinforcement learning tasks as effective as possible. Your goal is to write a reward function for the environment that will help the agent learn the task described in text. Your reward function should use useful variables from the environment as inputs. As an example, the reward function signature can be:

```
def compute_reward_shaped(obs: torch.Tensor, action: torch.
    Tensor) ->

 Tuple[[float, Dict[str, float]]]

    ...
    return reward, { }
```

the obs shape is {batch_size, obs_dim}and action shape is {batch_size, action_dim}. and batch_size is 1. Make sure any new tensor or variable you introduce is on the same device as the input tensors.

The Python environment is {task_obs_code_string}. Write a reward function for the following task: {task_description}.

The output of the reward function should consist of two items: (1) the total reward, (2) a dictionary of each individual reward component. The code output should be formatted as a python code string: "'''python ... '''".

Some helpful tips for writing the reward function code:

1. You may find it helpful to normalize the reward to a fixed range by applying transformations like `torch.exp` to the overall reward or its components.
2. If you choose to transform a reward component, then you must also introduce a temperature parameter inside the transformation function; this parameter must be a named variable in the reward function and it must not be an input variable. Each transformed reward component should have its own temperature variable.
3. Make sure the type of each input variable is correctly specified; a float input variable should not be specified as `torch.Tensor`.
4. Most importantly, the reward code's input variables must contain only attributes of the provided environment class definition (namely, variables that have the prefix `self.`). Under no circumstance can you introduce new input variables.

---

**Eureka Prompt 2: Reflection**

You are a reward engineer trying to write reward functions to solve reinforcement learning tasks as effective as possible. Your goal is to write a reward function for the environment that will help the agent learn the task described in text. Your reward function should use useful variables from the environment as inputs. As an example, the reward function signature can be:

```
def compute_reward_shaped(obs: torch.Tensor, action: torch.
    Tensor) -> Tuple[float, Dict[str, float]]:
    ...
    return reward, {}
```

the obs shape is {batch_size, obs_dim}and action shape is {batch_size, action_dim}. and batch_size is 1. Make sure any new tensor or variable you introduce is on the same device as the input tensors.

The Python environment is {task_obs_code_string}. Write a reward function for the following task: {task_description}.

The output of the reward function should consist of two items: (1) the total reward, (2) a dictionary of each individual reward component. The code output should be formatted as a python code string: "'''python ... '''".

Some helpful tips for writing the reward function code:

> 1. You may find it helpful to normalize the reward to a fixed range by applying transformations like `torch.exp` to the overall reward or its components.
>
> 2. If you choose to transform a reward component, then you must also introduce a temperature parameter inside the transformation function; this parameter must be a named variable in the reward function and it must not be an input variable. Each transformed reward component should have its own temperature variable.
>
> 3. Make sure the type of each input variable is correctly specified; a float input variable should not be specified as `torch.Tensor`.
>
> 4. Most importantly, the reward code's input variables must contain only attributes of the provided environment class definition (namely, variables that have the prefix `self.`). Under no circumstance can you introduce new input variables.
>
> {the best code}
> We trained a RL policy using the provided reward function code and tracked the values of the individual components in the reward function as well as global policy metrics such as success rates and episode lengths after every {epoch_freq} epochs and the maximum, mean, minimum values encountered:
> {data}
> Please carefully analyze the policy feedback and provide a new, improved reward function that can better solve the task. Some helpful tips for analyzing the policy feedback:
>
> 1. If the success rates are always near zero, then you must rewrite the entire reward function.
>
> 2. If the values for a certain reward component are near identical throughout, then this means RL is not able to optimize this component as it is written. You may consider:
>
>    (a) Changing its scale or the value of its temperature parameter,
>
>    (b) Re-writing the reward component,
>
>    (c) Discarding the reward component.
>
> 3. If some reward components' magnitude is significantly larger, then you must re-scale its value to a proper range.
>
> Please analyze each existing reward component in the suggested manner above first, and then write the reward function code.

Table 11: Average episodic returns for the task *drawer-open* across different evolution rounds ($r_x$) and agents ($a_x$), evaluated at 200k training steps. $r_x$ denotes the evolution round, and $a_x$ represents the agent in that round. The column *best* indicates the best-performing agent in each round.

| Round ($r_x$) | $a_1$ | $a_2$ | $a_3$ | $a_4$ | $a_5$ | $a_6$ | $a_7$ | $a_8$ | Best |
|---|---|---|---|---|---|---|---|---|---|
| $r_1$ | 1018.77 | 371.74 | 1931.21 | 2117.50 | 2145.50 | 2373.34 | 2483.56 | 1704.40 | $a_7$ |
| $r_2$ | 1964.78 | 357.45 | 1704.17 | 2268.05 | 1869.65 | 2073.90 | 2124.35 | 1561.83 | $a_4$ |
| $r_3$ | 2163.36 | 1123.12 | 801.96 | 1993.93 | 2163.62 | 1904.10 | 850.56 | 428.80 | $a_1$ |
| $r_4$ | 2142.97 | 839.31 | 1260.82 | 1445.53 | 1665.38 | 1470.83 | 433.12 | 1063.76 | $a_1$ |
| $r_5$ | 928.98 | 1392.44 | 1761.59 | 2123.26 | 2308.81 | 1348.77 | 698.31 | 1888.74 | $a_5$ |

## C   PROMPT DETAILS FOR THE Q-SHAPING FRAMEWORK

The Q-shaping framework necessitates a general template to guide the code generation provided by large language models (LLMs). This template requires three key components: (1) the code template, (2) the environment description, and (3) the environment configuration file.

Below is a comprehensive overview of the general template:

> **General Prompt**
>
> You need to generate a piece of code based on the description of the environment or the configuration file of the environment.
>
> The purpose of this code is to provide a suitable Q value for (s, a) that you consider good based on the information provided.For bad (s, a), you can assign a Q-value of 0 or a lower value to discourage the robot from taking this action.
> Requirements:
> 1. In short, your task is to convert the task description into a Python-style Q (s, a)
> 2. The environment description typically provides the obs_dim and action_dim, along with the conditions for terminal states and truncation. Your task is to penalize behaviors that lead to the end and encourage behaviors that result in high scores.
> 3. If you are confident, you can use your knowledge to generate (s,a,Q) values that you believe may lead to success or failure. 4. The code returns s, a, q_targets
> 5. Generate two functions, def good_Q(self, batch_size), def bad_Q(self, batch_size)
> 6.TIPS: Action is more important than state, so you should focus on encouraging actions that lead to success and discouraging actions that lead to failure.
> 7. When designing bad Q-values, there are no bad states, only bad actions. You need to clearly identify which state-action pairs lead to termination and avoid those actions.
> 8. If the description mentions states that lead to termination, you should include them in the bad Q-values, as assigning a Q-value of 0 to termination states usually accelerates learning.
> 9. You can try to encourage as many (s,a) pairs as possible to guide the agent to explore directions that you believe will lead to success.
> 10. You should provide a complete class definition, including the __init__, goodQ, and badQ methods, without omitting any of them.
>
> {code template}
>
> {environment description}
>
> {environment config file}

## C.1 ILLUSTRATIVE EXAMPLE: Q-SHAPING FRAMEWORK IN ACTION

To provide a concrete understanding of the Q-shaping framework, we present an example using the robotic arm task `"handle-press-v2"`. This example illustrates the application of the general template outlined earlier and demonstrates how the three key components—code template, environment description, and environment configuration file—come together to generate (s,a,Q) pairs that effectively guide agent behavior.

### C.1.1 ENVIRONMENT DESCRIPTION

The Meta-World benchmark is a suite of 50 diverse robotic manipulation tasks designed to evaluate reinforcement learning (RL) and meta-reinforcement learning (meta-RL) algorithms. In Yu et al. (2020), the authors introduce a simulated Sawyer robotic arm and provide detailed definitions of the observation space, action space, and evaluation metrics.

For the purpose of this paper, we focus on Section **4.1 Actions, Observations, and Rewards** from Yu et al. (2020), which outlines the design of the state space, action space, and reward functions. These details are critical for understanding how to guide large language models (LLMs) to generate high-quality $(s, a, Q)$ pairs.

> **Environment Description**
>
> **4.1 Actions, Observations, and Rewards**   In order to represent policies for multiple tasks with one model, the observation and action spaces must contain significant shared structure across tasks. All of our tasks are performed by a simulated Sawyer robot. The action space is a 2-tuple consisting of the change in 3D space of the end-effector followed by a normalized torque that the gripper fingers should apply. The actions in this space range between $-1$ and $1$. For all tasks, the robot must either manipulate one object with a variable goal position, or manipulate two objects with a fixed goal position. The observation space is represented as a 6-tuple of the 3D Cartesian positions of the end-effector, a normalized measurement of how open the gripper is, the 3D position of the first object, the quaternion of the first object, the 3D position of the second object, the quaternion of the second object, all of the previous measurements in the environment, and finally the 3D position of the goal. If there is no second object or the goal is not meant to be included in the observation, then the quantities corresponding to them are zeroed out. The observation space is always 39 dimensional.
>
> Designing reward functions for Meta-World requires two major considerations. First, to guarantee that our tasks are within the reach of current single-task reinforcement learning algorithms, which is a prerequisite for evaluating multi-task and meta-RL algorithms, we design well-shaped reward functions for each task that make each of the tasks at least individually solvable.
>
> More importantly, the reward functions must exhibit shared structure across tasks. Critically, even if the reward function admits the same optimal policy for multiple tasks, varying reward scales or structures can make the tasks appear completely distinct for the learning algorithm, masking their shared structure and leading to preferences for tasks with high-magnitude rewards.
>
> Accordingly, we adopt a structured, multi-component reward function for all tasks, which leads to effective policy learning for each of the task components. For instance, in a task that involves a combination of reaching, grasping, and placing an object, let $o \in \mathbb{R}^3$ be the object position, where $o = (o_x, o_y, o_z)$, $h \in \mathbb{R}^3$ be the position of the robot's gripper, $z_{\text{target}} \in \mathbb{R}$ be the target height of lifting the object, and $g \in \mathbb{R}^3$ be goal position. With the above definition, the multi-component reward function $R$ is the combination of a reaching reward, a grasping reward, and a placing reward or subsets thereof for simpler tasks that only involve reaching and/or pushing. With this design, the reward functions across all tasks have a similar magnitude that ranges between 0 and 10, where 10 always corresponds to the reward-function being solved, and conform to similar structure, as desired. The full form of the reward function and a list of all task rewards is provided in Appendix.

### C.1.2  Environment Configuration File

The primary purpose of the configuration file is to specify the target object's location and the initial position of the robotic arm's gripper. This information can assist the LLM in generating movement direction vectors that lead to effective actions.

```
1 from __future__ import annotations
2
3 from typing import Any
4
5 import numpy as np
6 import numpy.typing as npt
7 from gymnasium.spaces import Box
8
9 from metaworld.envs.asset_path_utils import full_v2_path_for
10 from metaworld.envs.mujoco.sawyer_xyz.sawyer_xyz_env import RenderMode,
    SawyerXYZEnv
11 from metaworld.envs.mujoco.utils import reward_utils
```

```python
from metaworld.types import InitConfigDict

class SawyerHandlePressEnvV2(SawyerXYZEnv):
    TARGET_RADIUS: float = 0.02

    def __init__(
        self,
        render_mode: RenderMode | None = None,
        camera_name: str | None = None,
        camera_id: int | None = None,
    ) -> None:
        hand_low = (-0.5, 0.40, 0.05)
        hand_high = (0.5, 1.0, 0.5)
        obj_low = (-0.1, 0.8, -0.001)
        obj_high = (0.1, 0.9, 0.001)
        goal_low = (-0.1, 0.55, 0.04)
        goal_high = (0.1, 0.70, 0.08)

        super().__init__(
            hand_low=hand_low,
            hand_high=hand_high,
            render_mode=render_mode,
            camera_name=camera_name,
            camera_id=camera_id,
        )

        self.init_config: InitConfigDict = {
            "obj_init_pos": np.array([0, 0.9, 0.0]),
            "hand_init_pos": np.array(
                (0, 0.6, 0.2),
            ),
        }
        self.goal = np.array([0, 0.8, 0.14])
        self.obj_init_pos = self.init_config["obj_init_pos"]
        self.hand_init_pos = self.init_config["hand_init_pos"]

        self._random_reset_space = Box(
            np.array(obj_low), np.array(obj_high), dtype=np.float64
        )
        self.goal_space = Box(np.array(goal_low), np.array(goal_high),
    dtype=np.float64)

    @property
    def model_name(self) -> str:
        return full_v2_path_for("sawyer_xyz/sawyer_handle_press.xml")

    @SawyerXYZEnv._Decorators.assert_task_is_set
    def evaluate_state(
        self, obs: npt.NDArray[np.float64], action: npt.NDArray[np.
    float32]
    ) -> tuple[float, dict[str, Any]]:
        (
            reward,
            tcp_to_obj,
            _,
            target_to_obj,
            object_grasped,
            in_place,
        ) = self.compute_reward(action, obs)

        info = {
            "success": float(target_to_obj <= self.TARGET_RADIUS),
            "near_object": float(tcp_to_obj <= 0.05),
            "grasp_success": 1.0,
```

```python
 75                 "grasp_reward": object_grasped,
 76                 "in_place_reward": in_place,
 77                 "obj_to_target": target_to_obj,
 78                 "unscaled_reward": reward,
 79             }
 80
 81             return reward, info
 82
 83         @property
 84         def _target_site_config(self) -> list[tuple[str, npt.NDArray[Any]]]:
 85             return []
 86
 87         def _get_pos_objects(self) -> npt.NDArray[Any]:
 88             return self._get_site_pos("handleStart")
 89
 90         def _get_quat_objects(self) -> npt.NDArray[Any]:
 91             return np.zeros(4)
 92
 93         def _set_obj_xyz(self, pos: npt.NDArray[Any]) -> None:
 94             qpos = self.data.qpos.flat.copy()
 95             qvel = self.data.qvel.flat.copy()
 96             qpos[9] = pos
 97             qvel[9] = 0
 98             self.set_state(qpos, qvel)
 99
100         def reset_model(self) -> npt.NDArray[np.float64]:
101             self._reset_hand()
102
103             self.obj_init_pos = self._get_state_rand_vec()
104             self.model.body("box").pos = self.obj_init_pos
105             self._set_obj_xyz(np.array(-0.001))
106             self._target_pos = self._get_site_pos("goalPress")
107             self.maxDist = np.abs(
108                 self.data.site("handleStart").xpos[-1] - self._target_pos[-1]
109             )
110             self.target_reward = 1000 * self.maxDist + 1000 * 2
111             self._handle_init_pos = self._get_pos_objects()
112
113             return self._get_obs()
114
115         def compute_reward(
116             self, actions: npt.NDArray[Any], obs: npt.NDArray[np.float64]
117         ) -> tuple[float, float, float, float, float, float]:
118             assert (
119                 self._target_pos is not None
120             ), "`reset_model()` must be called before `compute_reward()`."
121             del actions
122             obj = self._get_pos_objects()
123             tcp = self.tcp_center
124             target = self._target_pos.copy()
125
126             target_to_obj = obj[2] - target[2]
127             target_to_obj = np.linalg.norm(target_to_obj)
128             target_to_obj_init = self._handle_init_pos[2] - target[2]
129             target_to_obj_init = np.linalg.norm(target_to_obj_init)
130
131             in_place = reward_utils.tolerance(
132                 target_to_obj,
133                 bounds=(0, self.TARGET_RADIUS),
134                 margin=abs(target_to_obj_init - self.TARGET_RADIUS),
135                 sigmoid="long_tail",
136             )
137
138             handle_radius = 0.02
139             tcp_to_obj = float(np.linalg.norm(obj - tcp))
```

```
140        tcp_to_obj_init = np.linalg.norm(self._handle_init_pos - self.
       init_tcp)
141        reach = reward_utils.tolerance(
142            tcp_to_obj,
143            bounds=(0, handle_radius),
144            margin=abs(tcp_to_obj_init - handle_radius),
145            sigmoid="long_tail",
146        )
147        tcp_opened = 0
148        object_grasped = reach
149
150        reward = reward_utils.hamacher_product(reach, in_place)
151        reward = 1.0 if target_to_obj <= self.TARGET_RADIUS else reward
152        reward *= 10
153        return (reward, tcp_to_obj, tcp_opened, target_to_obj,
       object_grasped, in_place)
```

Listing 1: Config file for sawyer-handle-press-v2

### C.1.3 CODE TEMPLATE

In the code template section, a Python-style code snippet and its explanation are provided. The Python-style code defines the expected output format of the LLM, while the accompanying explanation helps the LLM better understand the structure and purpose of the code.

> **Code Template**
>
> For example:
> In the DrawerOpen environment, actions are encouraged when they involve moving the gripper towards the handle and closing the gripper. Specifically:
>
> Encouraged actions (good Q):
>
> Movement in the direction of the handle (positive y-direction).
> Closing the gripper, especially when the gripper is close to the handle.
> Discouraged actions (bad Q):
>
> Movement away from the handle (negative y-direction).
> Opening the gripper when it is near the handle, or further opening it when it's already open.
>
> ```
> 1  class DrawerOpen:
> 2      def __init__(self):
> 3          self.obs_dim = 39  # Observation space dimension
> 4          self.action_dim = 4  # Action space dimension (dx, dy,
>        dz, gripper torque)
> 5          self.maxDist = 0.2  # Maximum distance for drawer
>        opening
> 6          self.target_reward = 1000 * self.maxDist + 1000 * 2
> 7          self.close_gripper_threshold = 0.05  # Distance
>        threshold to encourage closing the gripper
> 8
> 9      def good_Q(self, batch_size):
> 10         actions = []
> 11         states = []
> 12         q_targets = []
> 13         for _ in range(batch_size):
> 14             # Generate a state where the gripper is approaching
>        the handle
> 15             handle_pos = np.array([0.0, 0.74, 0.09])  #
>        Approximate handle position
> ```

```
16          # Start gripper at a position slightly away from the
       handle
17          gripper_pos = handle_pos + np.random.uniform(-0.15,
       0.15, size=3)
18          gripper_open = np.random.uniform(0.0, 0.5)   #
       Gripper partially closed
19
20          # Construct the observation
21          obs = np.zeros(self.obs_dim)
22          obs[:3] = gripper_pos  # Gripper position
23          obs[3] = gripper_open  # Gripper state
24          obs[4:7] = handle_pos  # Handle position
25          obs[7:] = np.random.uniform(-0.1, 0.1, size=self.
       obs_dim - 7)   # Other observations
26
27          # Generate actions that move the gripper towards the
        handle (positive y movement)
28          direction_to_handle = handle_pos - gripper_pos
29          distance_to_handle = np.linalg.norm(
       direction_to_handle)
30          if distance_to_handle > 0:
31              action_direction = direction_to_handle /
       distance_to_handle
32          else:
33              action_direction = np.zeros(3)
34          action_magnitude = np.random.uniform(0.05, 0.1)
35          action_movement = action_direction *
       action_magnitude
36
37          # Encourage closing the gripper when close to the
       handle
38          if distance_to_handle < self.close_gripper_threshold
        :
39              gripper_action = np.random.uniform(0.5, 1.0)   #
       Close the gripper more aggressively
40          else:
41              gripper_action = np.random.uniform(0.0, 0.5)   #
       Keep the gripper partially open
42
43          action = np.concatenate((
44              action_movement,  # Move towards the handle
45              [gripper_action]  # Gripper action
46          ))
47
48          # Calculate a higher Q-value for actions that reduce
        the distance to the handle and close the gripper
49          new_gripper_pos = gripper_pos + action[:3]
50          new_distance_to_handle = np.linalg.norm(handle_pos -
        new_gripper_pos)
51          if new_distance_to_handle < self.
       close_gripper_threshold and gripper_action > 0.5:
52              q_value = (1.0 - new_distance_to_handle / self.
       maxDist) * 15.0  # Higher reward for closing near the handle
53          else:
54              q_value = max(0.0, 1.0 - new_distance_to_handle
       / self.maxDist) * 10.0
55
56          states.append(obs)
57          actions.append(action)
58          q_targets.append(q_value)
59
60      # Convert lists to tensors
61      states = torch.tensor(states, dtype=torch.float32)
```

```python
        actions = torch.tensor(actions, dtype=torch.float32).
    view(batch_size, self.action_dim)
        q_targets = torch.tensor(q_targets, dtype=torch.float32)
    .view(-1, 1)

        return states, actions, q_targets

    def bad_Q(self, batch_size):
        actions = []
        states = []
        q_targets = []
        for _ in range(batch_size):
            # Generate a state where the gripper is far from the
     handle
            gripper_pos = np.array([0.0, 0.5, 0.2]) + np.random.
    uniform(-0.1, 0.1, size=3)
            gripper_open = np.random.uniform(0.5, 1.0)  #
    Gripper open

            handle_pos = np.array([0.0, 0.74, 0.09])  # Handle
    position remains the same

            # Construct the observation
            obs = np.zeros(self.obs_dim)
            obs[:3] = gripper_pos  # Gripper position
            obs[3] = gripper_open  # Gripper state
            obs[4:7] = handle_pos  # Handle position
            obs[7:] = np.random.uniform(-1.0, 1.0, size=self.
    obs_dim - 7)  # Other observations

            # Generate actions that move the gripper away from
    the handle or open the gripper further
            # Positive y direction moves closer to the handle,
    so negative y moves away and should be punished
            action = np.array([
                np.random.uniform(-1.0, 1.0),      # Random x
    movement
                np.random.uniform(-1.0, 0.0),      # Negative y
    movement (away from handle)
                np.random.uniform(-1.0, 1.0),      # Random z
    movement
                np.random.uniform(-1.0, 0.0)      # Torque to
    open gripper
            ])

            # Assign a low Q-value for bad actions (moving away
    from the handle)
            q_value = 0.0

            states.append(obs)
            actions.append(action)
            q_targets.append(q_value)

        # Convert lists to tensors
        states = torch.tensor(states, dtype=torch.float32)
        actions = torch.tensor(actions, dtype=torch.float32).
    view(batch_size, self.action_dim)
        q_targets = torch.tensor(q_targets, dtype=torch.float32)
    .view(-1, 1)

        return states, actions, q_targets
```

Listing 2: Code template

In the example above, the `good_Q` method assigns high Q-values to actions likely to lead to success, whereas `bad_Q` discourages actions leading to termination. This approach aligns with the principles outlined in the general template.

### C.1.4 LLM OUTPUT

Based on the requirements and suggestions outlined in the `General Template`, as well as the observation and action space definitions provided in the `Environment Description` and `Configuration File`, and the `Code Template` discussed above, the LLM can generate good Q pairs and bad Q pairs that meet the specified requirements.

Below is the generated code:

```python
class HandlePress:
    def __init__(self):
        self.obs_dim = 39  # Observation space dimension
        self.action_dim = 4  # Action space dimension (dx, dy, dz,
        gripper torque)
        self.TARGET_RADIUS = 0.02  # Target radius for pressing the
        handle
        self.handle_init_pos = np.array([0.0, 0.9, 0.0])  # Approximate
        handle start position
        self.target_pos = np.array([0.0, 0.8, 0.14])  # Target position
        after pressing
        self.max_dist = np.abs(self.handle_init_pos[2] - self.target_pos
        [2])
        self.gripper_init_pos = np.array([0.0, 0.6, 0.2])  # Gripper
        initial position

    def good_Q(self, batch_size):
        actions = []
        states = []
        q_targets = []
        for _ in range(batch_size):
            # Generate gripper position near the handle
            gripper_pos = self.handle_init_pos + np.random.uniform(-0.8,
        0.8, size=3)
            # gripper_pos[2] += 0.1  # Ensure gripper is above the handle
         in z-axis
            gripper_open = np.random.uniform(-1, 1)  # Gripper partially
        closed

            # Construct the observation
            obs = np.zeros(self.obs_dim)
            obs[:3] = gripper_pos  # Gripper position
            obs[3] = gripper_open  # Gripper state
            obs[4:7] = self.handle_init_pos  # Handle position
            obs[7:] = np.random.uniform(-0.3, 0.3, size=self.obs_dim - 7)
          # Other observations

            # Generate actions that move the gripper towards the handle
        and press it down
            x_move = np.random.uniform(-0.001, 0.001)
            y_move = np.random.uniform(-0.001, 0.001)
            z_move = np.random.uniform(-0.1, -0.08)
            action_movement = np.array([x_move, y_move, z_move])
            # Apply torque to press the handle
            gripper_action = np.random.uniform(-0.001, 0.001)  # Apply
        torque to press

            action = np.concatenate((
                action_movement,  # Move towards the handle and press
        down
                [gripper_action]  # Gripper torque action
            ))
```

```
40
41                  # Calculate Q-value based on how much the handle is pressed
        towards the target
42                  handle_movement = action_movement[2]  # Z-axis movement (
        pressing down)
43                  new_handle_pos_z = self.handle_init_pos[2] + handle_movement
44                  handle_to_target = np.abs(new_handle_pos_z - self.target_pos
        [2])
45
46                  q_value = 10
47                  states.append(obs)
48                  actions.append(action)
49                  q_targets.append(q_value)
50
51          # Convert lists to tensors
52          states = torch.tensor(states, dtype=torch.float32)
53          actions = torch.tensor(actions, dtype=torch.float32).view(
        batch_size, self.action_dim)
54          q_targets = torch.tensor(q_targets, dtype=torch.float32).view(-1,
         1)
55
56          return states, actions, q_targets
57
58      def bad_Q(self, batch_size):
59          actions = []
60          states = []
61          q_targets = []
62          for _ in range(batch_size):
63              # Generate gripper position far from the handle
64              gripper_pos = self.gripper_init_pos + np.random.uniform(-0.8,
        0.8, size=3)
65              gripper_open = np.random.uniform(-1, 1)  # Gripper open
66
67              # Construct the observation
68              obs = np.zeros(self.obs_dim)
69              obs[:3] = gripper_pos  # Gripper position
70              obs[3] = gripper_open  # Gripper state
71              obs[4:7] = self.handle_init_pos  # Handle position
72              obs[7:] = np.random.uniform(-1.0, 1.0, size=self.obs_dim - 7)
         # Other observations
73
74              # Generate actions that move the gripper away from the handle
        or lift it up
75              action = np.array([
76                  np.random.uniform(-1.0, 1.0),  # Random x movement
77                  np.random.uniform(0.8, 1.0),  # Random y movement
78                  np.random.uniform(0.2, 1.0),  # Positive z movement (move
         up)
79                  np.random.uniform(-1.0, 1)  # Negative torque (release
        pressure)
80              ])
81
82              # Assign a low Q-value for bad actions
83              q_value = -10
84
85              states.append(obs)
86              actions.append(action)
87              q_targets.append(q_value)
88
89          # Convert lists to tensors
90          states = torch.tensor(states, dtype=torch.float32)
91          actions = torch.tensor(actions, dtype=torch.float32).view(
        batch_size, self.action_dim)
92          q_targets = torch.tensor(q_targets, dtype=torch.float32).view(-1,
         1)
```

```
93
94         return states, actions, q_targets
```

Listing 3: LLM generated code for HandlePress

## D LIMITATIONS AND DISCUSSION

In this section, we discuss several limitations of our work and propose promising directions for future research to address these challenges.

One limitation of the Q-shaping framework is its dependence on the understanding capabilities of large language models (LLMs). For tasks where the LLM cannot fully comprehend the dynamics of movement or the control-specific requirements of the task, the framework fails to deliver meaningful improvement. This limitation constrains the applicability of Q-shaping to domains where task requirements can be effectively interpreted by the LLM. Models with stronger reasoning capabilities may be needed to generate valid state-action pairs.

Another limitation is the difficulty of scaling Q-shaping to visual or real-world settings. The framework requires models capable of generating states, but current technology lacks models that can simultaneously process textual and visual inputs and output comprehensive state-action descriptions. This gap restricts the ability of Q-shaping to operate effectively in environments where visual data is a critical component. Future progress in multimodal modeling, such as vision-language models that integrate text and images, could alleviate this challenge by enabling richer state representations.

By addressing these limitations, Q-shaping has the potential to evolve into a more versatile framework capable of operating across diverse tasks and environments, ultimately advancing its impact on reinforcement learning research.

