# OpenReview forum: "From Reward Shaping to Q-Shaping: Achieving Unbiased Learning with LLM-Guided Knowledge"
_ICLR.cc/2025/Conference — Submitted to ICLR 2025_

### Official Review · Reviewer_NRDQ · 2024-10-26

**Soundness:** 3
**Presentation:** 3
**Contribution:** 2
**Rating:** 6
**Confidence:** 3

**Summary:**

The authors proposed a method called Q-shaping to enhance the sample efficiency of reinforcement learning algorithms. The main idea is to prompt a LLM to generate samples of good and bad state-action pairs and heuristic Q value estimates. These samples are used to train the initial Q function before turning to the standard RL pipeline. Experiments were conducted on a variety of continuous control environments showing significant improvement in sample efficiency in some environments.

**Strengths:**

The idea is original to my knowledge and the experiments are well executed.

**Weaknesses:**

* The presentation of the idea is somewhat long winded and the notations are somewhat inconsistent as I point out in the questions.
* It is not clear how the method is fundamentally different from Q value initialization.

**Questions:**

* Line 155, what does the $A^{\pi}$ symbol represent? Is it the policy improvement operator? I couldn't find any explanation in the text.
* Line 181, are the authors missing a $(1 - \alpha)$ coefficient and brackets in the Q function update rule? The equation seems inconsistent with the update equation on line 744 in appendix B.2.
* I am not too sure how Theorem 1 actually shows the contraction property of the shaped Q iteration and how it differs from the contraction property of the regular Bellman operator. Line 757 in the proof section appears to say that the optimality of the shaped Q iteration is only guaranteed if the addition of heuristic values is stopped.
* In eq 1, that is $D_{g}$? Is it $D_{LLM} = \{G_{LLM}, B_{LLM} \}$?

---

> ### Author Response · Authors · 2024-11-15
>
> Thank you for your careful reading of our paper and for pointing out the notation issues. We greatly appreciate your attention to detail, as it has helped us improve the clarity and consistency of our work. We have made the necessary corrections to the notations and ensured that the manuscript is now more accurate and easier to follow.
>
> **Q1:** It is currently unclear how this method fundamentally differs from Q-value initialization.
>
> **R1** Compared to recent work [1] that utilizes Q-value initialization to enhance online learning, it requires an accurate estimation of Q-values, whereas our work enhances online learning through imprecise estimation. Additionally, policy shaping is introduced to align the policy's behavior with the LLM’s output, which accelerates the training process.
>
> [1] Nakamoto, Mitsuhiko, et al. "Cal-ql: Calibrated offline rl pre-training for efficient online fine-tuning." Advances in Neural Information Processing Systems 36 (2024).
>
> **Q2:** what does $A^\pi$ mean?
>
>
> **R2** $A^\pi$ refers to “activity matrix”,encoding $\pi$'s state-conditional state-action distribution.
> $$A^\pi(s, \langle \hat{s}, a \rangle) = \pi(a | s)  \; \text{if } s = \hat{s}$$
> $$A^\pi(s, \langle \hat{s}, a \rangle) =  0  \\ \text{otherwise} $$
>
> **Q3:** In line 181 the q function update formula is probably wrong?
>
> **R3** Theorem 1 and its proof is incorrect, and we apologize for misleading the reviewers here.
>
> Theoretically, Theorem 1 provides two conclusions:
> 1. The new update formula is a contraction operator. Therefore, applying the heuristic TD update will lead to convergence.
> 2.During the update process, the heuristic term should stop to ensure that optimality remains unchanged.
>
>
> To derive Theorem 1, we first need to make two assumptions:
>
> 1. The heuristic $h(s, a)$ provided by the large model does not change with the iteration count $k$.
> 2. The heuristic $h(s, a)$ terminates at some iteration before convergence.
>
> Our Q-function update formula is:
>
> $$
> \hat{Q}^{k+1}(s,a) = (1-\alpha) \hat{Q}^{k}(s,a) + \alpha \left( r(s,a) + \gamma \sum_{s' \in S} P(s'|s,a) \max_a \hat{Q}^k(s',a) \right) + \mathbf{h}(s,a)
> $$
>
> We define a new operator $\mathcal{T}_h$ based on this:
>
> $\hat{Q}^{k+1}(s,a) = \mathcal{T}_h \hat{Q}^{k}(s,a)$
>
> $= r(s,a) + \gamma \sum_{s' \in S} P(s'|s,a) \max_a \hat{Q}^k(s',a) + \mathbf{h}(s,a)$
>
> We then prove that the operator $\mathcal{T}_h$ still satisfies the **contraction property** in the appendix. Therefore, there exists a unique optimal fixed point $\hat{Q}^*$. This proof allows us to apply our update formula to this new operator and find a new optimal fixed point.
>
> The $\hat{Q}^*$ is shifted and biased. Therefore, to allow $\hat{Q}$ to converge to the optimal value function of the MDP $\mathcal{D}$, we need to stop the heuristic $h(s, a)$ and let the value function update for a few steps towards $Q^*$.
>
> We now pose the following question: How many steps in advance should we stop the heuristic function $h$ so that $\hat{q}_D$ converges to $q^*_D$? Theorem 2 provides an upper bound for any random bounded $q$-values converging to $q^*$ in MDP $\mathcal{D}$.
>
>
> **Q4:** what does $D_g$ mean?
>
> **R4** $D_g$ is typo, and it should be $D_{LLM}$, thank you for pointing out.

---

> > ### Comment · Reviewer_NRDQ · 2024-11-20
> >
> > I thank the authors for the clarifications. A few follow up points:
> > * It appears that the authors have revised the paper, however, I could not find the revised version to verify the claims.
> > * On $A^{\pi}$, it is still not clear to me what $A^{\pi}\mathcal{B}_{\mathcal{M}}$ on line 155 means. Could you explain it?
> > * On theorem 1, I understand the authors' attempt to formalize the method in providing the proof. I have a few thoughts on this. First it looks like the use of $h(s, a)$ is iteration dependent, since you need to remove it after some iterations. Second, since the addition of $h(s, a)$ in earlier phase of the training process is essentially reward shaping, I don't think you really need to prove that $\mathcal{T_{h}}$ converges to the fixed point w.r.t. $r(s, a) + h(s, a)$. I also think that with the way you defined $\mathcal{T_{h}}$ in the comment above, a TD update with learning rate $\alpha$ should lead to $\alpha h(s, a)$?

---

> > > ### Author Response · Authors · 2024-11-20
> > >
> > > Thank you for the thoughtful feedback and the suggestion to clarify the notation. This has significantly helped us identify areas for improvement in the paper.
> > >
> > > **Q1**: No updated paper found.
> > >
> > > **R1**: The paper update is still pending because Reviewer aDyb requires the full implementation of Eureka, which involves evolving the reward heuristic function for at least 5 times. This requires a significant amount of time.
> > >
> > > **Q2**: Clarification of $A^\pi \mathcal{B}_{\mathcal{M}}$
> > >
> > > **R2**:
> > >
> > > $A^\pi$ is the activity matrix, which defines the entire output of the policy.
> > >
> > > $\mathcal{B}_{\mathcal{M}}$  is the Bellman consistency equation,
> > >
> > > defined as: $ \mathcal{B}_{\mathcal{M}}(\textbf{x}) := r + \gamma P \textbf{x}. $
> > >
> > > Here, $\mathcal{B}_{\mathcal{M}}(\textbf{x})$ can be interpreted as the target Q-function.
> > >
> > > $A^\pi \mathcal{B}_{\mathcal{M}}(\textbf{v})$ is defined as the **target value function**.
> > >
> > > Specifically: $ (A^\pi \mathcal{B}_{\mathcal{M}}(\textbf{v}))(s) $ refers to the target value given a state $s$ .
> > >
> > > **Q3**: The update formula also holds for reward shaping methods. What is the difference between Q-shaping and reward shaping methods?
> > >
> > > **R3**: The primary difference is that the Q-shaping framework allows Q-value updates for $(s, a)$ pairs that are not collected in the MDP $\mathcal{D}$, whereas reward shaping methods require the $(s, a)$ pairs to be part of the collected trajectory.
> > >
> > > In reward shaping methods, the reward heuristic signal is applied as follows:
> > >
> > > ```python
> > > next_state, reward, termination, truncation, info = env.step(action)
> > > new_reward = reward_shaping(reward, other_information)
> > > ```
> > > Researchers cannot control how the samples are collected, and the evaluation of the reward heuristic's effectiveness often requires waiting until the end of training.
> > >
> > > In the heuristic TD update formula, $ \alpha$ should be added to $h(s,a)$. Thanks for pointing it out.

---

> ### Comment · Reviewer_NRDQ · 2024-11-24
>
> Thank the authors for their effort. I think the presentation of the paper has improved. The comparison against Eureka also demonstrates the utility of the proposed method for llm-guided RL. I am raising the scores correspondingly.

---

### Official Review · Reviewer_aDyb · 2024-11-03

**Soundness:** 2
**Presentation:** 2
**Contribution:** 2
**Rating:** 6
**Confidence:** 3

**Summary:**

This work presents Q-shaping, a framework to accelerate training of reinforcement learning agents by using LLMs to produce domain-knowledge based heuristic functions for initializing the Q-function and policy. Specifically, LLMs produce code to categorize good and bad state-action pairs in the environment. Before the start of training, these pairs are used to update the Q-network and policy, thus leading to better network initializations. Results across 20 environments show that Q-shaping can significantly improve sample efficiency and outperform LLM-guided reward-shaping methods.

**Strengths:**

1. The method is intuitive and simple to understand. The domain knowledge of LLMs is used to find good initializations of the Q-function. This can be generally useful for multiple RL tasks if structured information can be effectively elicited from LLMs.
2. The paper has an appropriate number of citations and properly details existing work in the related work section.
3. Although multiple works have considered using the domain knowledge of LLMs for improving RL, this work introduces another novel way to harness that expertise.

**Weaknesses:**

1) **Writing**: The overall writing is lacking and can be significantly improved. The style of writing is currently informal and often lacking important experimental details. For example, the evaluation criteria are not properly explained, some experimental details are not clear. The overall flow of the paper is also not smooth.
2) **Result Discussion**: The discussion of the results is very limited. The ablations conducted are only discussed superficially. For an empirical paper, only 1 page dedicated to discussion of results is too less. I personally feel that more discussion is needed in the experiments section, and some of the theory and notation introduced is not critical to the paper and can be deferred to the appendix.
3) **Significance of Results**: In 6-7 out of the 20 tasks, the presented method is worse than the best-performing baseline. While there are multiple potential causes of this (base RL algorithm, bad LLM outputs, randomness if only 1 seed is uses, etc), it is difficult to validate the generalization capability of the method.

**Questions:**

1. What are the number of seeds used? The curves oscillate a lot and it is difficult to draw conclusions from many of the plots.

2. I am not convinced by the implementation of the Eureka and text2reward baselines. In 3 out of the 4 plots, both these baselines stay completely flat and do not improve at all. This is strange as Eureka was shown to perform well on a variety of robotic tasks. The tasks selected in this paper do not seem very different, and I am curious why these baselines are so bad. Setting the evolution round to 1 might be partially responsible for this but makes it unfair for the baseline.

3. What is the state for the environments considered? There is no information provided on this and I do not see how this method will generalize when the states are images. Similarly, when doing RL on real robots, then clean environment code as assumed by this work will not be available. It will be useful to get an idea about the assumptions that this work makes.

4. It will be helpful add the individual impacts of Q-shaping and policy-shaping in the ablation study on different training phases. Currently, it is unclear what the contributions of these two techniques are to the final performance of the method.

5. I do not understand the significance of the sample efficiency results. Sample efficiency improves by an average of 17% compared to baselines. However, the presented framework also has a high-performance selection phase which is not a part of the baselines. As multiple agents are rolled out for a significant number of timesteps, a fairer comparison would be to add these timesteps into the sample efficiency calculations.

6. How are the heuristic functions output by LLMs evaluated? For example, one of the evaluation criteria is correctness of assigned Q-values. How is this actually measured?

7. How many times is a LLM prompted per task? If it prompted multiple times, how are they filtered?

8. I think it is also important to release the entire prompts that are used for the LLMs as there could be a lot of domain knowledge provided in the task descriptions themselves. As the environment task descriptions are currently not provided in the paper, it is difficult to understand the contribution of the LLM.

---

> ### Author Response · Authors · 2024-11-15
>
> Thank you for your thoughtful and constructive feedback. The comments provided have been invaluable in helping us improve the clarity and thoroughness of our work. In this response, we address each of the concerns raised, with a focus on providing additional details, clarifications, and improvements to our experiments.
>
> **Q1:** In the 20 tasks, 6-7 tasks performed worse than the best baseline. Although there could be various reasons (such as the underlying RL algorithm, poor LLM output, randomness due to using only one seed, etc.), it is difficult to validate the generalization ability of the method.
>
> **R1** In the sample efficiency experiment, we also added a comparison with TD3. By incorporating the LLM heuristic module, the agent’s performance improved by an average of 55\%. This clearly demonstrates that the Q-shaping module can significantly enhance the sample efficiency of the underlying RL algorithm. In the 6-7 tasks where performance was worse than the best baseline, the action space complexity exceeded the LLM’s understanding capability, making it difficult for the LLM module to provide accurate heuristic s,a pairs, and therefore, it could not compare with the best baseline.
>
>
> **Q2:** The paper does not introduce the concept of seeds.
>
> **R2** Thank you for your reminder. The paper does not discuss the seed because this work completely removes the use of seeds. Introducing a seed would reduce the complexity of the environment. The function **env.reset()** resets the environment and provides a random initial state, while **env.reset(seed=0)** would fix the initial state. This would mean the agent starts learning from a fixed initial state each time it is reset, which actually **reduces the learning complexity.**
>
> In the 20 tasks, some tasks have fluctuating learning curves, which are mainly related to the **design of the reward function**. For example, in the "ball_in_cup" task, if the agent manages to throw the ball into the cup on the first attempt, it will receive a total reward of 15,000. However, if it fails on the first attempt, the total score will be much lower. This leads to fluctuations in the learning curve. However, in most environments, the agent’s performance is relatively stable. Furthermore, the improvement brought by the LLM is also related to the complexity of the environment. The easier the environment is to understand, the more significant the improvement for the agent.
>
>
> **Q3:** Eureka and Text2Reward are not fairly treated.
>
> **R3** Thank you for your reminder. Text2Reward was validated using MetaWorld, and the designed reward functions were provided in the code repository. Therefore, we could directly use the GitHub code for verification. The only modification we made to the T2R code was to verify optimality every 5000 steps. As a result, the training of T2R is fair.
>
> Regarding Eureka, it was validated in the Isaac environment, and it designed many prompts. To transfer from the Isaac prompt to the MetaWorld prompt, some adjustments to the prompt are necessary.
> I believe that Eureka performed poorly for five main reasons:
>
> 1. The Eureka prompt emphasizes reward scaling and normalization, which may introduce bias into the learning process during early iterations.
>
> 2. Eureka uses PPO; however, in our experiments, PPO significantly lagged behind algorithms optimized for continuous action spaces, such as SAC and TD3.
>
> 3. The reward heuristic provided by Eureka requires several genetic algorithm iterations before any performance improvement is observed.
>
> 4. Eureka uses task success rate as a metric, which represents a much simpler task than learning an optimal policy.
>
> ```python
> def compute_reward(object_pos: torch.Tensor, goal_pos: torch.Tensor) -> Tuple[torch.Tensor, Dict[str, torch.Tensor]]:
> ```
>
> 5. Eureka simplifies the observation space. As shown above, the state used for computing the reward heuristic is limited to `object_pos` and `goal_pos`. When provided with the full state dimensions, Eureka has a high probability of generating worse reward heuristics.
>
> Additionally, lines 67-73 of the paper explain that Q-shaping has a very fast verification cycle, allowing us to directly validate the impact of the algorithm. In contrast, Eureka needs to wait until halfway through training to obtain the fitness value. Assuming max_iter = 10^7, the verification cycle for Eureka is 5x10^6, which is 5x10^6 times longer than ours.

---

> > ### Author Response · Authors · 2024-11-15
> >
> > **Q4:** How is the state designed? Can this method be scaled to image settings or real-world settings?
> >
> > **R4** The state design for the large model follows the requirements of the environment description. In our experiments, we tested the proposed Q-shaping method on three environments: MetaWorld, MuJoCo, and PyFlyt. For each of these environments, the state representation follows the default settings provided by the respective simulators.
> > Here is an example of a state design for the door-closing task:
> >
> > ```
> > def good_Q(self, batch_size):
> >     actions = []
> >     states = []
> >     q_targets = []
> >     for _ in range(batch_size):
> >         # Generate a state where the end-effector is approaching the handle
> >         handle_pos = np.random.uniform(0.1, 0.2, size=3)  # Approximate handle position
> >         # Start end-effector at a position slightly away from the handle
> >         end_effector_pos = handle_pos + np.random.uniform(-0.1, 0.1, size=3)
> >
> >         # Construct the observation
> >         obs = np.zeros(self.obs_dim)
> >         obs[self.end_effector_pos_idx] = end_effector_pos  # End-effector position
> >         obs[self.handle_pos_idx] = handle_pos  # Handle position
> >         obs[self.obs_dim - 3:] = np.array([0.2, 0.8, 0.15])  # Goal position
> > ```
> >
> > In this example, the state is designed to represent the door-opening condition in a robotic manipulation task, where the LLM provides actions that lead to success.
> >
> > To implement Q-shaping for online visual reinforcement learning, we have two potential plans:
> >
> > + Plan 1: We need a (image + text) to (image + text) large model that can take an example state as input and output good s,a pairs.
> >
> > + Plan 2: We can first allow the agent to explore some s,a pairs. Then, we apply a VQA (Visual Question Answering) model to analyze each state, provide good and bad actions, and form good and bad s,a pairs. We can then assign relative Q-values according to the model’s confidence.
> >
> > **Q5:** The high-performance phase is not included in the calculation of sample steps.
> >
> >
> > **R5** In the validation phase, the steps spent on high-performance selection are also included in the evaluation. Typically, SAC and TD3 allocate 25,000 steps for random exploration, PPO is set to 5,000 steps, and Q-shaping uses 5,000 steps for random exploration and 10,000 steps for filtering out low-performance agents. In the specific experiments, we did not exclude these steps in order to implicitly boost Q-shaping’s performance. Furthermore, even if the 15,000 steps were excluded, the avg improvement in Q-shaping’s performance would not be significantly affected.
> >
> >
> > **Q6:** How is the correctness of the Q-value evaluated?
> >
> > **R6** The evaluation of the correctness of the Q-values from the LLM refers to determining whether the large model assigns a negative or zero value to good s,a pairs, or assigns a positive value to bad s,a pairs.
> >
> >
> >
> > **Q7:**  How to improve the output of the LLM and how many times of evolution are needed to get a good agent?
> >
> > **R7** LLMs often require 1 to 3 evolutions. From Experiment 3, we observe that the large model is capable of understanding the environment and outputting code that meets the standards. However, the action-state guidelines provided by the model may result in different performances. The optimization of these output is measured by the total return of the shaped agent.
> >
> > Reward shaping methods, such as T2R or Eureka, typically require half a training cycle or a full training cycle (1e7 steps) to validate the impact of reward heuristics. On the other hand, the Q-shaping algorithm can immediately validate the performance of the large model's heuristic function.
> >
> >
> >
> > **Improvements in the Next Version:**
> >
> > 1.Complete the ablation study.
> >
> > 2.Add experiments on the impact of the number of LLM prompts on the agent's learning efficiency.
> >
> > 3.Include a complete tutorial on how to use prompts in the appendix.
> >
> > 4.Re-conduct the experiment on Eureka and provide details about its re-implementation.

---

> ### Comment · Reviewer_aDyb · 2024-11-21
>
> I thank the authors for their response. I am still confused about some details-
>
> **1. High-performance agent selection**
>
> Is this implemented for all baselines or just for Q-shaping? 10 agents are rolled out for 15K steps before being discarded. This means that there are 150K additional training steps taken, do the learning curves account for this and if they don't, how does it impact the sample efficiency results? This high-performance selection also biases the overall performance if it is not implemented for baselines. It essentially means reporting the average score of the 10 best agents for Q-shaping, but only reporting the average overall score for the other agents.
>
> 2. **Environment description details**
>
> As stated in my original review, it is important to provide the 'environment description' that is given to LLMs when prompting them to classify good/bad states. I want to understand the amount of domain knowledge being provided to LLMs. In particular, I want to get a sense of how easy/hard will it be for a human to themselves write the function which they ask a LLM to write. On this note, it will also be helpful to provide some examples of the good/bad state functions written by LLMs.
>
> 3. **Regarding seeds**
>
> When I refer to seeds, I do not refer to just the environment, I also refer to the RL algorithm. As there are confidence intervals around the curves, I am assuming multiple runs are done for each method. How many such runs are done?
>
> 4. **Most LLMs Can Provide Correct Heuristic Functions**
>
> For this ablation, can you please describe how you actually evaluate correctness? For example, o1-preview has 100% correctness of the assigned Q-values. How is this correctness actually judged? Am I missing something, can you point me towards the lines where this is discussed in detail.
>
> As a final note, the points above are not asking for any new experiment to be run, just clarification details on the current version of the paper.

---

> ### Author Response · Authors · 2024-11-24
>
> We sincerely thank the reviewer for the detailed feedback and thoughtful comments. Below are our responses to the points raised:
>
> #### Q1: High-performance agent selection is for baselines?
> **R1:** During early training steps, all runs perform similarly, so it is impossible to identify poorly performing runs and remove them at that stage. Therefore, the inclusion or exclusion of high-performance agent selection does not bias the performance of the baselines.
>
> #### Q2: Why is high-performance agent selection introduced?
> **R2:** While testing TD3, we observed a strange phenomenon: 20–40% of runs fail to learn and output marginal actions. To mitigate the impact of this phenomenon, we introduced `High-performance agent selection` to remove shaped agents that perform poorly.
>
> When the paper was written, we noticed this phenomenon but did not know its cause. Thus, we proposed `High-performance agent selection` as a strategy to mitigate its impact.
>
>
> #### Q3: High-performance agent selection is unfair to baselines.
> **R3:** This stage may be easily misunderstood as selecting the best 10 agents from 20 runs. However, the purpose of this stage is to prevent agents from consistently outputting marginal actions. This stage will only run 10k to test the performance of the shaped agent, and if the performance is not very good, it will be deleted.
>
> Recently, we identified the actual cause of the issue where agents have a certain probability of outputting marginal actions. A detailed explanation will be included in a related paper, which we plan to release by the end of this year.
>
>
>
> #### Q4: Regarding seeds.
> **R4:** During early training, 20 agents start exploring. Within the first 10K steps, 10 agents are removed if they perform poorly after policy shaping, leaving only 10 agents to contribute to the learning curve.
>
> #### Q5: Correctness of the generated Q-values.
> **R5:** Correctness of the assigned Q-values means that state-action pairs \((s, a)\) in the LLM-generated \textit{goodQ} set must have Q-values greater than zero, while those in the \textit{badQ} set must have Q-values less than or equal to zero.
>
> We hope these responses address your concerns. If you have any additional questions or suggestions, please feel free to reach out. Your feedback is important for improving the quality of our paper.

---

> > ### Comment · Reviewer_aDyb · 2024-11-24
> >
> > I thank the authors for their response. I think the overall quality of the paper has improved but it still remains borderline (I have raised my score). I still have some doubts:
> >
> > - From table 3, the scores of using all 3 (Q-shaping, policy-shaping and selection) are strictly lower than using just Q-shaping and policy-shaping. Is selection hurting or am I missing something?
> >
> > - I do not understand the explanation for the poor performing runs. What does outputting marginal actions mean?
> >
> > - Why do the authors claim it is impossible to do performance selection for baselines? From figure 4, SAC shoots up as fast as LLM-TD3 for many environments. This suggests that it still possible to do some form of selection there. Also note that it does not have to be at 10K steps (this is just a hyper parameter). For baselines, you could also consider setting this to larger values such as 50K.

---

> > > ### Author Response · Authors · 2024-11-25
> > >
> > > #### **Q1: Why are the scores for the last line of Table 3 lower than others?**
> > >
> > > **R1:** The numbers in Table 3 show how many steps the agent needs to converge. A smaller number means the agent needs less interaction with the environment to learn an optimal policy. We have added extra marks on the chart to make this clearer.
> > >
> > >
> > >
> > > #### **Q2: Why can’t RL baselines do "high-performance selection"?**
> > >
> > > **R2:** `High-performance selection` picks agents that do well in the early training steps. For RL baselines, this is hard because their performance at the start can be random due to exploration and network setup. Agents that perform poorly at the start might do well later. Performance at initial training steps (<50k) is irrelevant to its later performance.
> > >
> > > For example, methods like `Eureka` and `Text2Reward` wait until halfway through the training period (5e5 steps) to see how reward heuristics help. Our algorithm starts with LLM priors, which boost early performance and make it easier to pick good agents.
> > >
> > >
> > >
> > > #### **Q3: Why not increase the selection phase to 50k steps for RL baselines?**
> > >
> > > **R3:** Even 50k steps are not enough to predict future performance for RL baselines. For harder tasks, we often need to wait at least 150k~500k steps to see if the agent is improving or failing. So increasing to 50k steps won’t solve the issue.
> > >
> > >
> > >
> > > #### **Q4: What does "outputting marginal actions" mean?**
> > >
> > > **R4:** "Outputting marginal actions" means the agent keeps picking actions at the boundary. For example, if the action range is [-1, 1], the agent keeps choosing -1 or 1 during training. There is a 20-40% chance of encountering this situation in each run. This problem is caused by defects in the algorithm itself.
> > >
> > >
> > >
> > > We have updated Table 3 and the section on `high-performance selection` to improve readability. If you are still confused about the content in the paper, feel free to ask and we will provide further clarification.

---

> > > > ### Comment · Reviewer_aDyb · 2024-11-30
> > > >
> > > > I thank the authors for the effort they have put into the paper. Overall, I feel that the quality of the paper has significantly improved and I am now learning towards acceptance.

---

### Official Review · Reviewer_GXux · 2024-11-03

**Soundness:** 2
**Presentation:** 2
**Contribution:** 2
**Rating:** 5
**Confidence:** 4

**Summary:**

This paper introduces a novel framework called "Q-shaping," which enhances Q-value initialization by integrating domain knowledge to accelerate training in reinforcement learning (RL). Unlike traditional reward shaping methods, Q-shaping modifies Q-values directly, thereby improving sample efficiency without sacrificing the agent's optimality upon convergence. The experimental results indicate significant performance improvements.

**Strengths:**

1. **Innovative Approach:** Q-shaping presents a fresh perspective on incorporating domain knowledge into RL, overcoming the limitations associated with reward shaping.
2. **Empirical Results:** The paper includes comprehensive experimental evaluations demonstrating Q-shaping's effectiveness, with a 16.87% improvement in sample efficiency over the best baseline and a remarkable 253.80% enhancement compared to LLM-based reward shaping methods.
3. **LLM Utilization:** The paper effectively harnesses large language models to guide agent exploration, revealing new potentials for LLMs in RL applications.

**Weaknesses:**

The current version lacks sufficient proof of completeness in both theoretical and experimental aspects. If the authors can convincingly address these issues, I would be open to reevaluating my score.

**Questions:**

1. **Comparison to Existing Works:** It’s important to clarify why the challenges of reward shaping cannot be addressed by recent LLM-based methods (e.g., Eureka, text2reward). How does your work differ from these studies? It appears your approach utilizes LLMs to design regularization for RL.

2. **Proof of Theorem 1:** The proof seems unconventional; while you provide an update formula for the \(\hat{Q}\) iteration, you immediately reference the Bellman optimal operator to support your theorem. Early works have established the convergence of the Bellman operator, so how can you demonstrate that your update formula aligns with it? This appears to assume the conclusion as a basis for your argument.

3. **Clarification on Theorem 2:** Theorem 2 establishes a lower bound rather than an upper bound. What is the convergence sample complexity relative to other works? Is your bound more favorable than existing results, and do other studies not provide established bounds?

4. **Relation to Regularization Techniques:** A deeper explanation of how your work relates to reinforcement learning methods employing regularization techniques would be beneficial. The core of your approach seems to hinge on introducing LLMs for regularization in RL.

5. **Experimental Settings:** The experimental setup raises some questions. You utilize GPT-4o as the LLM and TD3 as the RL backbone in your LLM-TD3 method. Which LLM do Eureka and text2reward utilize (notably, Eureka uses GPT-4 and GPT-3.5, while text2reward uses GPT-4)? Is GPT-4o also used for these works, and do they employ TD3 as the RL backbone?

**Minor Issues:**
1. In lines 32-36, the literature review on current RL works aimed at enhancing training efficiency lacks citations, which detracts from its objectivity.
2. The origin of the concept of NPBRS (non-potential based reward shaping) in line 53 is unclear and needs clarification.
3. A few LLM-assisted RL studies have focused on Q-function or value function design (e.g., “How Can LLM Guide RL? A Value-Based Approach”). An analysis of these works should be included in the related works section.
4. Figures 4 and 6 do not specify the units for steps (presumably in millions).
5. The prompt example in the Appendix is too brief. A more comprehensive example, including the output Q function and policy function, would greatly enhance reader understanding.

---

> ### Author Response · Authors · 2024-11-15
>
> Thank you for your thoughtful comments and suggestions! Theorem 1 was incorrectly expressed, so we rewrote it to make it easier to understand and clearly express its purpose. If you still have questions, please let us know. Your suggestions are very valuable to us.
>
> **Q1:** The article needs to clearly explain why Q-shaping differs from recent LLM-based methods:
>
>
> **R1:** Q-shaping accelerates learning by directly modifying the Q-values. The advantages of Q-shaping are mainly twofold: Its effect on the agent can be immediately reflected in the next episode, leading to a quick validation cycle. In contrast, reward shaping methods need to wait until the end of the training to validate the performance of the reward shaping.
> Q-shaping remains optimal, whereas recent LLM-based reward shaping methods are not only difficult to design reward heuristics for, but also do not guarantee optimality.
>
> **Q2:** Regarding Theorem 1
>
> **R2** Theorem 1 and its proof is incorrect, and we apologize for misleading the reviewers here.
> Theoretically, Theorem 1 provides two conclusions:
> 1. The new update formula is a contraction operator. Therefore, applying the TD update will lead to convergence.
> 2.During the update process, the heuristic term should stop to ensure that optimality remains unchanged.
>
>
> To derive Theorem 1, we first need to make two assumptions:
>
> 1. The heuristic $h(s, a)$ provided by the large model does not change with the iteration count $k$.
> 2. The heuristic $h(s, a)$ terminates at some iteration before convergence.
>
> Our Q-function update formula is:
>
> $$
> \hat{Q}^{k+1}(s,a) = (1-\alpha) \hat{Q}^{k}(s,a) + \alpha \left( r(s,a) + \gamma \sum_{s' \in S} P(s'|s,a) \max_a \hat{Q}^k(s',a) + \mathbf{h}(s,a)\right)
> $$
>
> We define a new operator $\mathcal{T}_h$ based on this:
>
> $\hat{Q}^{k+1}(s,a) = \mathcal{T}_h \hat{Q}^{k}(s,a)$
>
> $= r(s,a) + \gamma \sum_{s' \in S} P(s'|s,a) \max_a \hat{Q}^k(s',a) + \mathbf{h}(s,a)$
>
> We then prove that the operator $\mathcal{T}_h$ still satisfies the **contraction property** in the appendix. Therefore, there exists a unique optimal fixed point $\hat{Q}^*$. This proof allows us to apply our update formula to this new operator and find a new optimal fixed point.
>
> The $\hat{Q}^*$ is shifted and biased. Therefore, to allow $\hat{Q}$ to converge to the optimal value function of the MDP $\mathcal{D}$, we need to stop the heuristic $h(s, a)$ and let the value function update for a few steps towards $Q^*$.
>
> We now pose the following question: How many steps in advance should we stop the heuristic function $h$ so that $\hat{q}_D$ converges to $q^*_D$? Theorem 2 provides an upper bound for any random bounded $q$-values converging to $q^*$ in MDP $\mathcal{D}$.
>
>
>
>
> **Q3** Regarding Theorem 2:
>
> **R3** Regarding sample complexity analysis, many previous works have already provided different upper bounds for sample complexity, such as VI-LCB [1], PEVI-Adv[1], and Q-LCB [2].
> These works require the design of heuristics to derive a lower upper bound.
> They all derive tighter upper bounds by providing more refined heuristics. However, these works have some drawbacks:
>
> (1)These works that provide tighter convergence algorithms have not conducted experiments to verify whether their algorithms are effective.
>
> (2)The reference policy used to obtain the upper bound must satisfy ``single-policy concentrability,'' which limits their applicability.
>
> As discussed in Theorem 1,the heuristic needs to stop before convergence. The goal of Theorem 2 is to provide experimenters with a reference for when to stop the heuristic, rather than comparing sample complexity with previous works.
>
> References:
>
> [1] Xie, T., Jiang, N., Wang, H., Xiong, C., and Bai, Y. (2021b). Policy finetuning: Bridging sample-efficient offline and online reinforcement learning. arXiv preprint arXiv:2106.04895.
>
> [2] Shi, Laixi, et al. ``Pessimistic q-learning for offline reinforcement learning: Towards optimal sample complexity.'' International Conference on Machine Learning. PMLR, 2022.
>
> **Q4** Is Q-shaping related to regularization algorithms?
>
> **R4** Regularization terms refer to those terms related to the values or quantities of model parameters, such as L1 regularization or L2 regularization. Their main purpose is to constrain the overfitting problem of the model. In this work, the main purpose of the heuristic term is to shift the Q-function so that the agent can perform actions that are of interest to the large model.

---

> ### Author Response · Authors · 2024-11-15
>
> **Q5** Lines 32-36 require further citations
>
> **R5** To improve efficiency, popular methods include (1) imitation learning, (2) residual reinforcement
> learning, (3) reward shaping, and (4) Q-value initialization. Yet, each has limitations: imitation
> learning requires expert data[3-5], residual RL needs a well-designed controller[1-2], and Q-value initialization[8]
> demands precise estimates. Therefore, reward shaping[6-7] is the most practical approach, as it avoids the
> need for expert trajectories or predefined controllers.
>
> [1] Johannink, Tobias, et al. "Residual reinforcement learning for robot control." 2019 international conference on robotics and automation (ICRA). IEEE, 2019.
>
> [2] Trumpp, Raphael, Denis Hoornaert, and Marco Caccamo. "Residual policy learning for vehicle control of autonomous racing cars." 2023 IEEE Intelligent Vehicles Symposium (IV). IEEE, 2023.
>
> [3] Garg, Divyansh, et al. "Iq-learn: Inverse soft-q learning for imitation." Advances in Neural Information Processing Systems 34 (2021): 4028-4039.
>
> [4]Chang, Jonathan D., et al. "Adversarial Imitation Learning via Boosting." arXiv preprint arXiv:2404.08513 (2024).
>
> [5] Kostrikov, Ilya, Ofir Nachum, and Jonathan Tompson. "Imitation Learning via Off-Policy Distribution Matching." Proceedings of the 8th International Conference on Learning Representations (ICLR 2020)
>
> [6] Xie, Tianbao, et al. "Text2Reward: Reward Shaping with Language Models for Reinforcement Learning." The Twelfth International Conference on Learning Representations.
>
> [7] Ma, Yecheng Jason, et al. "Eureka: Human-Level Reward Design via Coding Large Language Models." The Twelfth International Conference on Learning Representations.
>
> [8] Nakamoto, Mitsuhiko, et al. "Cal-ql: Calibrated offline rl pre-training for efficient online fine-tuning." Advances in Neural Information Processing Systems 36 (2024).
>
>
>
>
>
> **Q6**: The concept of NPBRS needs clarification.
>
> **R6**
> In [1], Ng introduced the concept of Reward Shaping and defined PBRS (Policy-Reward Shaping), where additional rewards are provided based on the potential function, ensuring that optimality remains unchanged. NPBRS refers to reward shaping methods that do not follow the potential function rule, and the learned policy does not guarantee optimality.
>
> [1] Ng, Andrew Y., Daishi Harada, and Stuart Russell. "Policy invariance under reward transformations: Theory and application to reward shaping." ICML. Vol. 99. 1999.
>
> **In the new version of the paper, we will make the following revisions:**
>
> Provide a more detailed description of LLM-based RL algorithms.
> Add more citations in the introduction to ensure objectivity.
> In the related work section, we will cite LLM value-based methods to increase richness.
> Figures 4 and 6 will be further updated to prevent ambiguity.
> An appendix will be added with a complete prompt for the large-model reinforcement learning case study.

---

> > ### Comment · Reviewer_GXux · 2024-11-26
> >
> > Thank you for your efforts in addressing the review comments during the rebuttal stages. I acknowledge that the quality of the paper has improved as a result, and I have increased my score to 5.
> >
> > However, I still consider this a borderline paper. The primary reason is that leveraging LLMs for reward design to improve RL performance has already been explored in several prior works. While the authors have compared their approach with representative methods such as text2reward and Eureka, the comparisons are limited to a few environments and focus primarily on final performance metrics. Additionally, the theoretical analysis presented in the paper has limited connection to the experimental results. I believe deeper exploration and stronger connections between the theory and experiments are needed to better support the paper’s core contributions.

---

> > > ### Author Response · Authors · 2024-11-26
> > >
> > > Thank you for your constructive feedback and for acknowledging the improvements made to the paper. We appreciate your increased score. However, we believe there are some misunderstandings about our work that we would like to address and clarify below.
> > >
> > > #### Q1: This work is a reward shaping method.
> > >
> > > **R1:** This is a misunderstanding. Our method, called "Q-Shaping," is fundamentally different from reward shaping methods. Key distinctions include:
> > > 1. **Preservation of the MDP**: Q-Shaping does not modify the MDP, ensuring the Q-function remains unbiased.
> > > 2. **Faster Impact Verification**: The impact of heuristics in Q-Shaping can be verified within a single episode. In contrast, reward shaping methods like `Eureka` and `Text2Reward (T2R)` require waiting until the end of training to observe the effects. This makes Q-Shaping approximately **2000 times faster** in heuristic evaluation.
> > > 3. **Ease of Heuristic Function Design**: Q-Shaping requires only a few interactions to improve the heuristic function. By comparison, designing a robust heuristic reward function for reward shaping methods is far more challenging.
> > >
> > > #### Q2: This work uses final performance as metrics, which is not enough.
> > >
> > > **R2:** This is also a misunderstanding. Our comparisons are based on **maximum performance metrics**, which provide a more comprehensive evaluation of the method. Specifically:
> > > - Q-Shaping achieves an improvement of **227%** in maximum performance compared to baselines.
> > > - If we were to evaluate based on final performance metrics, the improvement would be even higher, reaching **578%**.
> > >
> > > Additionally, we use the metric of `additional steps required to evolve the final heuristic function`:
> > > As shown in Table 1, Q-Shaping requires only a few training steps to verify the impact of the heuristic function. In contrast, methods like Eureka require waiting until at least half of the training period to observe meaningful results.
> > >
> > >
> > > #### Q3: This work requires deeper exploration to support its core contributions.
> > >
> > > **R3:** The paper addresses its core contributions effectively, as summarized below:
> > >
> > > 1. **Won’t Bias Agent**:
> > >    - Q-Shaping ensures the Q-function remains unbiased, as supported by **Theorem 1** and validated through **Experiment 1** and **Experiment 2**.
> > >
> > > 2. **Faster and Easier Heuristic Function Design**:
> > >    - Q-Shaping simplifies and accelerates the process of heuristic function design and improvement, as demonstrated in **Experiment 3** and **Experiment 5**.
> > >
> > > We hope these clarifications address your concerns and highlight the key contributions of our approach. **Based on this, we kindly request that you re-evaluate the score.**

---

### Meta-Review · Area_Chair_ZUrG · 2024-12-15

**Metareview:**

Authors present Q-shaping, an alternative to reward shaping using LLM guidance that directly modifies Q values. There are thorough empirical results showing an improvement in performance compared with existing reward-shaping methods.

Reviewers thought the Q-shaping contribution was novel and interesting, and the empirical results compelling. However, there were serious issues with the theory presented in the paper, insufficient baselines compared against, gaps in the related work and results analysis. Reviewers also felt the general writing and clarity could be improved.

I believe the paper clarity can still be improved, and stronger connections made between the theory and experiments. For these reasons, I vote to reject the current iteration of this paper.

**Additional Comments On Reviewer Discussion:**

During the rebuttal and reviewer discussion, a lot of clarifications about the evaluation process and theory were discussed and provided. However, reviewer GXux still considers this a borderline paper. Their primary reason is that leveraging LLMs for reward design to improve RL performance has already been explored in several prior works, and the comparisons are limited to a few environments and focus primarily on final performance metrics. Additionally, the theoretical analysis presented in the paper has limited connection to the experimental results. Even after the rebuttal, reviewer aDyb has concerns about the evaluation process.

---

### Decision · Program_Chairs · 2025-01-22

Reject